# Estimating the true stability of the prehydrolytic outward-facing state in an ABC protein

**Márton A Simon**[1,2,3]**, Iordan Iordanov**[1,2,3]**, Andras Szollosi**[1,2,3]**, László Csanády**[1,2,3]*****

[1]Department of Biochemistry, Semmelweis University, Budapest, Hungary; [2]HCEMM-SE Molecular Channelopathies Research Group, Budapest, Hungary; [3]HUN-REN-SE Ion Channel Research Group, Budapest, Hungary

**Abstract** CFTR, the anion channel mutated in cystic fibrosis patients, is a model ABC protein whose ATP-driven conformational cycle is observable at single-molecule level in patch-clamp recordings. Bursts of CFTR pore openings are coupled to tight dimerization of its two nucleotide-binding domains (NBDs) and in wild-type (WT) channels are mostly terminated by ATP hydrolysis. The slow rate of non-hydrolytic closure – which determines how tightly bursts and ATP hydrolysis are coupled – is unknown, as burst durations of catalytic site mutants span a range of ~200-fold. Here, we show that Walker A mutation K1250A, Walker B mutation D1370N, and catalytic glutamate mutations E1371S and E1371Q all completely disrupt ATP hydrolysis. True non-hydrolytic closing rate of WT CFTR approximates that of K1250A and E1371S. That rate is slowed ~15-fold in E1371Q by a non-native inter-NBD H-bond, and accelerated ~15-fold in D1370N. These findings uncover unique features of the NBD interface in human CFTR.

## Editor's evaluation

This important study provides a convincing mechanistic dissection of the contributions to gating and ATP hydrolysis from different CFTR channel mutations that have been exploited in structural studies to stabilize the open state of the channel. This is achieved by comparing structural information between channel orthologues from different species, by performing single-channel and macroscopic current recordings and cleverly combining different residue substitutions, and calculating mutant cycles. The precise quantitation of the effects of mutations in this study allows estimation of the energetics of a key transition in the channel gating cycle that had remained elusive. The findings are important for biophysicists and physiologists interested in CFTR channels and ABC transporters in general.

*For correspondence:
csanady.laszlo@med.
semmelweis-univ.hu

## Introduction

The CFTR anion channel is a key component of transepithelial salt-water transport in the lung, the pancreas and the intestine, and its mutations cause cystic fibrosis (*O'Sullivan and Freedman, 2009*). CFTR (ABCC7) is the only known ion channel member of the family of ATP-Binding Cassette (ABC) proteins. Its two ABC-typical halves, each consisting of a transmembrane domain (TMD1 and 2, *Figure 1A*, *gray*) followed by a cytosolic nucleotide-binding domain (NBD1 and 2; *Figure 1A*, *blue* and *green*), are linked by a unique regulatory (R) region (*Figure 1A*, *pale red*) which must be phosphorylated by cAMP-dependent protein kinase (PKA) to allow channel activity (*Riordan et al., 1989*; *Cheng et al., 1991*).

**Figure 1.** CFTR topology and gating cycle. (**A**) Cartoon topology of CFTR. TMD1–2, *gray*; NBD1, *blue*; NBD2, *green*; R-domain, *pale red*. Red asterisk denotes catalytic site mutations. (**B**) Cartoon of head-to-tail NBD dimer. Color coding as in (**A**). ATP, large yellow circles. Site 2 (*top*): conserved Walker A/B residues, *small red circles*; catalytic base, *small black circle*; signature sequence, *magenta crescent*. Site 1 (*bottom*): color coding as for site 2, *pale residues* represent non-conserved substitutions. Residue numbering corresponds to the human CFTR sequence. (**C**) Schematic gating cycle of phosphorylated CFTR. Flickery closures from states $B_1$ and $B_2$ are not depicted. Disruption of ATP hydrolysis in catalytic site mutants (*red cross*) reduces gating to reversible $IB_1 \leftrightarrow B_1$ transitions (*red box*). Note, that in reality sites 1 and 2 are equally near the membrane, in the cartoons site 2 is depicted as the top and site 1 as the bottom site merely for representation purposes.

ABC proteins are present in all kingdoms of life. The 48 human ABC proteins play essential physiological roles by mediating transmembrane transport of a variety of substrates (*Dean et al., 2022*). Within the ABC family the NBDs are the most highly conserved modules, and consist of a core subdomain that binds ATP (the 'head') and an ABC-specific alpha-helical subdomain (the 'tail'). In all ABC proteins in the presence of ATP the two NBDs form a tight head-to-tail dimer which occludes two ATP molecules (*Figure 1B*, *yellow*) at the interface. Both nucleotide-binding sites are flanked on one side by the conserved Walker A and Walker B motifs in the head of one NBD (*Figure 1B*, *red*), and on the other side by the conserved 'ABC signature sequence' in the tail of the other NBD (*Figure 1B*, *magenta*). The ATP-bound NBD dimer is stable, but is disrupted following ATP hydrolysis (*Locher, 2016*). In a subset of ABC proteins, which includes the entire C subfamily, only one of these two

composite ATP-binding sites is catalytically active (*Procko et al., 2009*). In CFTR the composite site formed by the head of NBD2 and the tail of NBD1 ('site 2', *Figure 1B*, *upper site*) contains canonical sequence motifs and hydrolyzes ATP with an overall turnover number of ~0.5–1 s$^{-1}$ (*Li et al., 1996*; *Liu et al., 2017*). In contrast, the other site ('site 1', *Figure 1B*, *lower site*) has accumulated non-canonical substitutions in several key motifs (*Figure 1B*, *pale residues*) and is catalytically inactive (*Aleksandrov et al., 2002*; *Basso et al., 2003*).

During substrate transport the TMDs of ABC exporters alternate between inward-facing (IF) and outward-facing (OF) conformations. In a phosphorylated CFTR channel opening and closure of the transmembrane ion pore (gating) follows analogous TMD movements (*Csanády et al., 2019*; *Figure 1C*). CFTR likely adopts an IF conformation, in which the pore is sealed near its extracellular end, during the long lived (~1 s) state (*Figure 1C*, *left*) which corresponds to long 'interburst' (IB) closed dwell times observable in single channel current recordings. We refer to these closed states as 'IF states'. When CFTR adopts the OF conformation the external gate is predominantly open, and a lateral portal which connects the channel pore with the cytosol generates a transmembrane aqueous pathway permeable to anions (*Zhang et al., 2018*; *Figure 1C*, *right*, *double arrows*). However, functional studies show that the continuity of the transmembrane pore is occasionally disrupted for brief (~10 ms) intervals by a smaller conformational change (not depicted in *Figure 1C*) likely confined to the external ends of the TMD helices (*Zhang et al., 2017*; *Zhang et al., 2018*; *Simon and Csanády, 2021*). Correspondingly, in single-channel recordings, this 'OF' or 'bursting' (B) state corresponds to clusters of channel openings separated by brief (flickery) closures (*Winter et al., 1994*). The OF state is also relatively stable, with a dwell time of hundreds of milliseconds.

In ABC exporters NBD and TMD movements are coupled: formation of the tight ATP-bound NBD dimer flips the TMDs from an IF to an OF conformation, and disruption of the tight NBD dimer following ATP hydrolysis resets the TMDs to the IF conformation (*Locher, 2016*). Gating of phosphorylated CFTR channels is driven by an analogous unidirectional conformational cycle (*Csanády et al., 2019*; *Figure 1C*). In all ABCC proteins, the OF state of the TMDs is coupled to tight dimerization of both NBD catalytic sites (*Johnson and Chen, 2018*; *Zhang et al., 2018*; *Huang et al., 2023*). Correspondingly, functional studies on CFTR showed that site 2 is tightly dimerized in the B state (*Vergani et al., 2005*; *Figure 1C*, *right*). In contrast, in IF structures of ABCC proteins obtained in the absence of ATP (*Johnson and Chen, 2017*; *Liu et al., 2017*; *Huang et al., 2023*), or for unphosphorylated CFTR even in its presence (*Levring et al., 2023*), the NBDs are seen to widely separate losing all contact across both composite sites. Interestingly, in the asymmetric bacterial ABC protein Tm287–288 some contacts across the degenerate site are retained throughout the entire conformational cycle, even in the IF state, as shown by extensive structural (*Hohl et al., 2012*; *Hohl et al., 2014*) and functional (*Timachi et al., 2017*) studies. Similarly, for phosphorylated CFTR channels functional studies suggested that site 1 does not completely separate throughout the entire ATP-driven gating cycle, even during IB events (*Tsai et al., 2010*; *Szollosi et al., 2011*) (see NBDs in *Figure 1C*, *left*), a prediction confirmed by recent single-molecule FRET measurements (*Levring et al., 2023*).

In ABC exporters thermodynamically uphill transport requires unidirectional conformational cycling. The 'coupling ratio' (CR), that is, the fraction of initiated cyles that are completed through ATP hydrolysis, depends on the relative rates of the two possible exit pathways from the prehydrolytic OF state. In most ABC transporters the actual values of these rates are hard to directly estimate, but in CFTR direct measurements of conformational dwell times are made feasible by single-channel current recordings, providing estimates for microscopic transition rates. Thus, for a wild-type (WT) channel ATP hydrolysis (rate $k_1$, *Figure 1C*) vs. non-hydrolytic NBD dimer dissociation rate ($k_{-1}$, *Figure 1C*) can be estimated. Because $k_1 \gg k_{-1}$, CR = $k_1/(k_1 + k_{-1})$ is near unity. In particular, for WT CFTR the mean burst duration ($\tau_b$) reports the sum of the life times of the prehydrolytic (*Figure 1C*, state $B_1$) and posthydrolytic (*Figure 1C*, state $B_2$) OF conformations. Because the life time of $B_2$ is short compared to that of $B_1$ ($k_2 \gg k_1$; *Figure 1C*; *Gunderson and Kopito, 1995*; *Csanády et al., 2010*), $1/\tau_b$ provides a rough estimate of rate $k_1$, which is ~4–5 s$^{-1}$ at room temperature for pre-phosphorylated CFTR channels gating in ATP. In contrast, rate $k_{-1}$ in WT CFTR is difficult to directly assess, and its estimates are based on the burst-state stability of various catalytic site mutants (*Figure 1A*, *red star*) in which ATP hydrolysis is disrupted (*Figure 1C*, *red cross*) and gating in saturating ATP is thus reduced to reversible $IB_1 \leftrightarrow B_1$ transitions (*Figure 1B*, *red box*). Numerous studies in the past have employed multiple different mutations for that purpose, including NBD2 Walker A lysine mutation K1250A (*Gunderson*

*and Kopito, 1995*; *Carson et al., 1995*; *Zeltwanger et al., 1999*; *Vergani et al., 2003*; *Csanády et al., 2006*; *Csanády et al., 2013*; *Csanády and Töröcsik, 2014*), Walker B aspartate mutation D1370N (*Gunderson and Kopito, 1995*; *Bompadre et al., 2005*; *Csanády et al., 2010*; *Sorum et al., 2015*; *Yeh et al., 2015*; *Sorum et al., 2017*; *Simon and Csanády, 2021*), and E1371S/Q mutations which eliminate the catalytic base (*Vergani et al., 2003*; *Bompadre et al., 2005*; *Zhou et al., 2005*; *Vergani et al., 2005*; *Zhou et al., 2006*; *Csanády et al., 2013*; *Yu et al., 2016*). Of note, so far all OF structures of CFTR (PDBID: 6msm, 6o2p, 6o1v, 7svd, 7sv7), as well as OF structures of several other ABC proteins (6bhu, 6cov, 6hbu, 6s7p, 7ekl, 8iza), were obtained from E-to-Q catalytic glutamate mutants.

However, interpretation of such mutant data is hampered by the fact that – even when compared under identical experimental conditions (expression system, temperature, and phosphorylation state) – non-hydrolytic closing rates of the above mutants scatter over a range of ~200-fold, from ~0.0025 $s^{-1}$ for E1371Q (*Vergani et al., 2005*; *Yu et al., 2016*) to ~0.5–1 $s^{-1}$ for D1370N (*Vergani et al., 2003*; *Yeh et al., 2015*). The reason for that large scatter is unknown. One extreme possibility is that ATP hydrolysis is completely abolished only in the slowest closing mutant (E1371Q) which therefore faithfully reports WT $B_1$-state stability, whereas in the faster closing mutants significant residual hydrolytic activity persists. Such a scenario would call into question conclusions based on the assumption of equilibrium gating in the latter mutants (*Sorum et al., 2015*; *Yeh et al., 2015*; *Sorum et al., 2017*; *Simon and Csanády, 2021*). The other extreme possibility is that all of the above mutations eliminate ATP hydrolysis, but different mutations differentially affect – either increase or decrease – $B_1$-state stability. In either case, estimating the true stability of state $B_1$ (i.e., rate $k_{-1}$) for WT CFTR would substantially promote our understanding of gating energetics. The aim of the present study was to estimate the true rate $k_{-1}$ for a pre-phosphorylated WT CFTR channel gating in ATP, as well as to verify disruption of ATP hydrolysis and evaluate effects on $B_1$-state stability of the various non-hydrolytic mutations that have been extensively employed as models in the past. The results suggest unique features of the site 2 NBD interface in human CFTR compared to other ABCC family proteins.

## Results
### Interfacial H-bond between NBD1 D-loop and the mutated NBD2 catalytic glutamate side chain suggested by structures of OF human, but not zebrafish, E-to-Q CFTR

The life time of the $B_1$ state of non-hydrolytic CFTR mutants can be conveniently measured in inside-out macro-patch recordings following activation of pre-phosphorylated CFTR channels by a brief exposure to ATP: the time constant of the macroscopic current relaxation following ATP removal reports $\tau_b$. A previous study that compared NBD dimer stabilities of the human and zebrafish orthologs of CFTR (hCFTR and zCFTR) uncovered a puzzling discrepancy between the two CFTR variants. Whereas for hCFTR a large difference between the burst durations of the serine vs. glutamine mutant of the catalytic glutamate (hE1371S vs. hE1371Q) had long been documented (cf., *Figure 2A*, *black* vs. *dark blue trace*; *Figure 2B*, *black* vs. *dark blue symbols*), no significant difference (p = 0.092) between $\tau_b$ of the analogous mutants of the zebrafish ortholog (zE1372S vs. zE1372Q; *Figure 2A*, *brown* vs. *light blue trace*; *Figure 2B*, *brown* vs. *light blue symbols*) was detectable (*Simon and Csanády, 2023*).

Those unexpected findings suggested differences in the fine structure of the site 2 interface in the bursting state of the two orthologs. We therefore examined the OF cryoelectron microscopy (cryo-EM) structures of the E-to-Q mutants of both orthologs (*Figure 2C, D*) in more detail. In hCFTR (PDBID: 6msm) close examination identified a hydrogen bond across the site 2 interface, formed between the amide nitrogen of the engineered hQ1371 side chain and the peptide carbonyl oxygen of residue hG576, located in the opposing D-loop of NBD1 (*Figure 2E*, *purple dotted line*; cf., electron density in *Figure 2—figure supplement 1*). That H-bond cannot be present in WT CFTR, as the native hE1371 side chain contains no chemical moiety suitable to form a stabilizing interaction with the opposing carbonyl group. Interestingly, in the zCFTR structure (PDBID: 5w81) the amide and carbonyl groups of the zQ1372 side chain are swapped. Thus, although residue hG576 aligns with zT575, the zQ1372 nitrogen points toward the backbone carbonyl group of zH576 and the distance between the two groups is substantially larger (4.3 vs. 3.0 Å), precluding formation of an H-bond (*Figure 2F*, *light purple dotted line*). Moreover, even if that swapped assignment of the zQ1372 side chain was

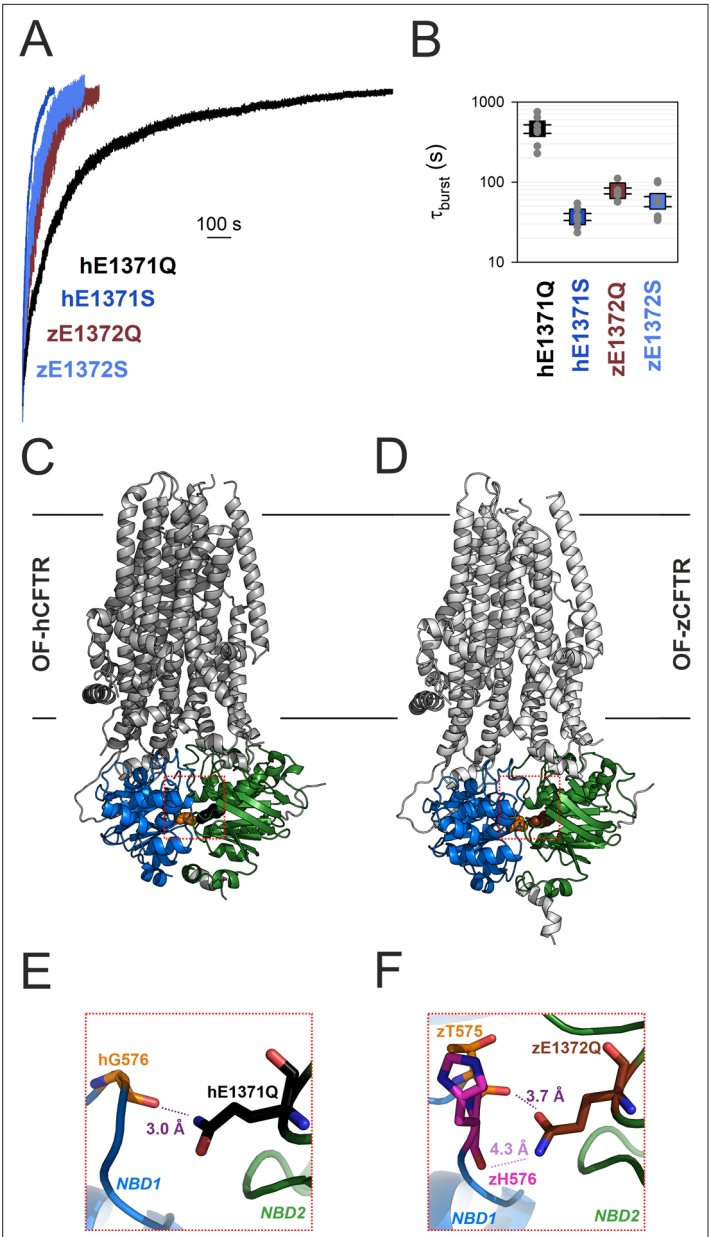

**Figure 2.** Non-native inter-NBD H-bond in OF E-to-Q mutant hCFTR, but not zCFTR, suggested by cryo-EM structures. (**A**) Macroscopic current relaxations upon ATP removal in inside-out patches excised from *Xenopus laevis* oocytes expressing hE1371Q (*black trace*), hE1371S (*dark blue trace*), zE1372Q (*brown trace*), and zE1372S (*light blue trace*) CFTR channels. Currents were activated by exposure to saturating ATP, following prior phosphorylation for ~2 min with 300 nM bovine PKA catalytic subunit. Current amplitudes are shown normalized by their steady-state values in ATP (i.e., just before ATP removal), membrane potential ($V_m$) was −20 to −80 mV. (**B**) Mean burst durations (in seconds) obtained as the time constants of single-exponential fits to the macroscopic relaxations shown in (**A**). Data are shown as mean ± standard error of the mean (SEM) from six to nine experiments using a logarithmic ordinate. (**A**, **B**) has been adapted from Figures 2A, B, and 3C, D from **Simon and Csanády, 2023**. (**C**, **D**) Cryo-EM structures of hCFTR-E1371Q (PDBID: 6msm) and zCFTR-E1372Q (PDBID: 5w81) in the OF conformation. Color coding as in (**A**), positions hG576/zT575 (*orange*), hQ1371 (*black*), and zQ1372 (*brown*) are shown in spacefill. *Red dotted squares* identify regions expanded in panels (**E, F**). (**E, F**) Close-up views of the regions surrounding the mutated catalytic glutamate side chains (*black/brown sticks*) in OF hCFTR (**E**) and zCFTR (**F**).

The online version of this article includes the following figure supplement(s) for figure 2:

**Figure supplement 1.** Experimental electron densities around residues G576 and Q1371 in OF structure of hCFTR E1371Q.

incorrect, the zQ1372-zT575 distance would still be larger (3.7 Å, *Figure 2F*, *dark purple dotted line*) than in hCFTR. These observations raised the possibility that the >10-fold longer life time of the bursting state of hE1371Q compared to hE1371S CFTR (*Figure 2A, B*) reflects artificial stabilization of state $B_1$ by the introduced non-native interfacial H-bond, which is absent in the zebrafish channel.

## In E1371Q-hCFTR the hQ1371 side-chain amino and hG576 peptide carbonyl groups are energetically coupled in the bursting state

Gating state-dependent changes in energetic coupling between two protein positions can be experimentally verified and quantified using thermodynamic mutant cycle analysis (*Vergani et al., 2005*). If the two target positions indeed interact, and the strength of the interaction changes during gating, then the kinetic effects of disrupting that interaction by single mutations at either position will not be additive in the double mutant. Hypothesized interactions between amino acid side chains can be conveniently perturbed by changing the length or chemical nature of the participating side chains. Perturbing a backbone carbonyl group is less straightforward, but if the target position is located in a loop, then shortening the loop by a single-residue deletion might increase the distance to the partner position sufficiently to disrupt the hypothesized interaction (*Simon and Csanády, 2021*).

Following the above strategy we verified the effect on burst-state stability of disrupting the hypothesized hG576–hQ1371 H-bond in the human E1371Q CFTR background construct (*Figure 2E*). To perturb position 1371, we shortened the side chain by substituting the glutamine with a serine (hE1371S). To perturb the position of the opposing backbone carbonyl group, we shortened the D-loop by deleting residue G576 (hG576Δ). We first addressed the effects of these perturbations on non-hydrolytic burst duration in macroscopic current relaxation experiments (*Figure 3A*). Whereas in the hE1371Q background deletion of residue hG576 decreased $\tau_b$ by ~20-fold (*Figure 3A*, *orange vs. black trace*; *Figure 3B*, *orange vs. black symbols*), in the hE1371S background the effect on $\tau_b$ of the same deletion was only ~2-fold (*Figure 3A*, *green vs. blue trace*; *Figure 3B*, *green vs. blue symbols*). From an energetic point of view, the mutation-induced shortening of $\tau_b$ reflects destabilization of the $B_1$ state relative to the transition state for non-hydrolytic closure ($T^{\ddagger}$). The resulting reduction in the free enthalpy barrier for non-hydrolytic closure ($\Delta\Delta G^0_{T^{\ddagger}-B1}$; *Figure 3C*, *numbers on arrows*) is quantified by the logarithm of the fractional change in $\tau_b$ (see Materials and methods). The coupling energy ($\Delta\Delta G_{int}(B_1 \rightarrow T^{\ddagger})$; *Figure 3C*, *purple*) is a measure of the discrepancy in the effects caused by perturbation of one target position depending on whether the other target position is mutated or intact (*Figure 3C*, *parallel arrows*). $\Delta\Delta G_{int}(B_1 \rightarrow T^{\ddagger})$ is thus given by the difference between $\Delta\Delta G^0_{T^{\ddagger}-B1}$ values on parallel sides of the cycle (*Figure 3C*, *numbers on parallel arrows*). $\Delta\Delta G_{int}(B_1 \rightarrow T^{\ddagger})$ thus quantifies the change in the strength of the hG576–hQ1371 interaction in the hE1371Q background construct while the channel proceeds from the $B_1$ state to state $T^{\ddagger}$ (see Materials and methods). That $\Delta\Delta G_{int}(B_1 \rightarrow T^{\ddagger})$ is significantly different from zero (p = 4.6 × 10$^{-5}$), its magnitude (2.3 ± 0.3 kT) is comparable to that of an H-bond, and its positive signature reports that the bond is present in the $B_1$ state, but not in state $T^{\ddagger}$. Because in the IB state separation of the NBD interfaces of site 2 (*Vergani et al., 2005*) renders formation of the inter-NBD H-bond unlikely, these results suggest that the bond is present selectively in the B state.

## In E1371Q-hCFTR the hQ1371–hG576 H-bond is maintained in the flickery closed state

The bursting state is a composite state which comprises the open (O) and the flickery closed ($C_f$) state. The equilibrium between those two states is reflected by the fraction of time the pore spends open within a burst ($P_{o|B}$), which is related to the intraburst equilibrium constant through the equation $K_{eq|B} = P_{o|B}/(1 - P_{o|B})$. Intraburst gating of non-hydrolytic CFTR mutants can be conveniently studied in inside-out patches that contain small numbers of channels, through dwell-time analysis of segments of record following ATP removal in which all but one channel have terminally closed. Due to the absence of ATP such 'last-channel' time windows (*Figure 3D*, *black bars*) are devoid of IB events, and allow selective collection of large numbers of open and flickery closed events (*Simon and Csanády, 2021*).

To address a potential change in the strength of the hG576–hQ1371 H-bond between the open and the flickery closed state of E1371Q hCFTR, we compared intraburst gating of hE1371Q and of the single and double target-site mutants (*Figure 3D*). In contrast to the large effects of the mutations on $\tau_b$, the high intraburst $P_o$ of the hE1371Q background construct ($P_{o|B} = 0.91 \pm 0.1$ ($n = 5$)) was not

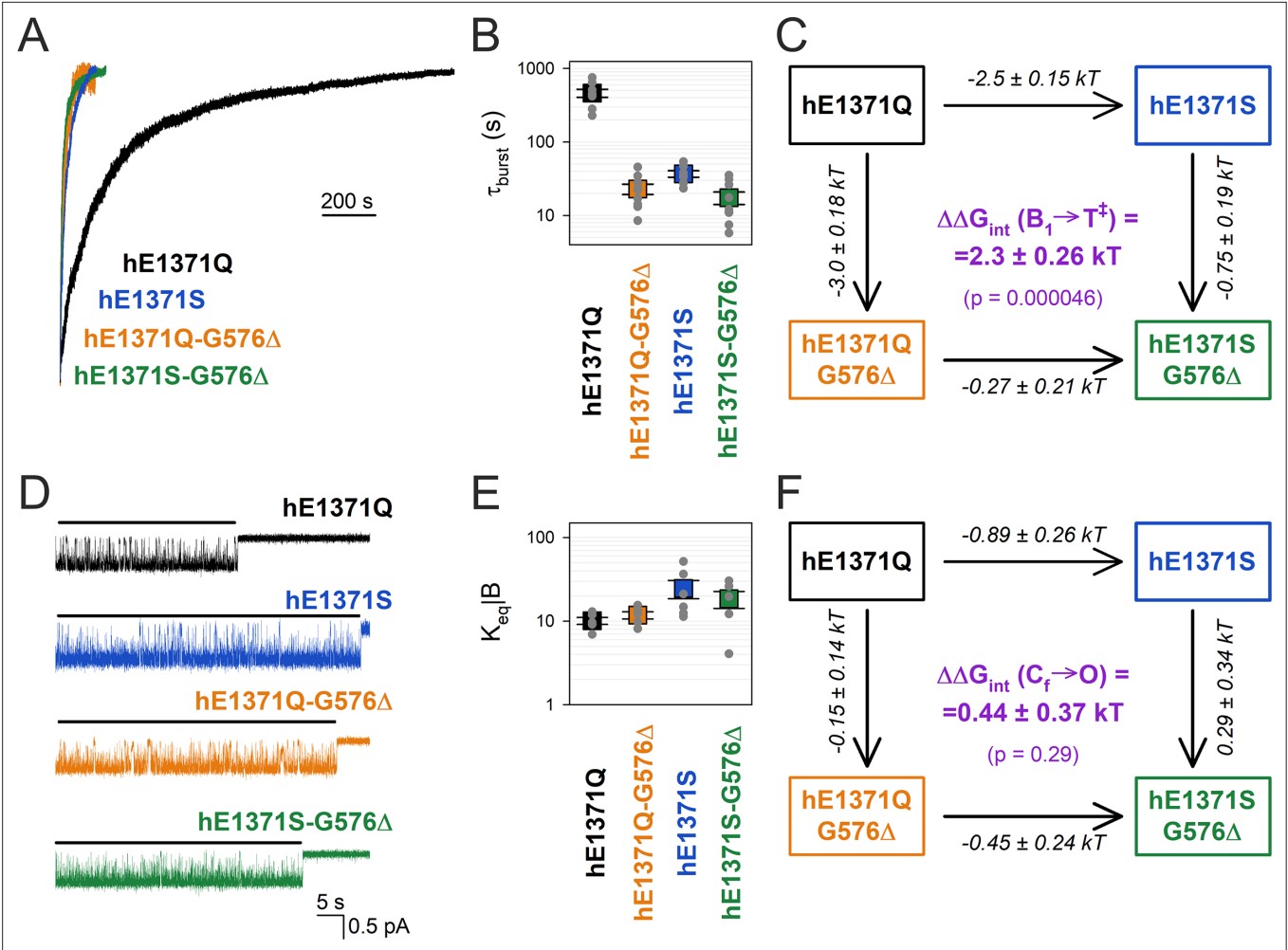

**Figure 3.** Strong coupling between positions 576 and 1371 of hCFTR E1371Q in the bursting state. (**A**) Macroscopic current relaxations following ATP removal for indicated human CFTR channel mutants (*color coded*). Inside-out patch currents were activated by exposure of pre-phosphorylated channels to saturating ATP. Current amplitudes are shown normalized by their steady-state values in ATP (i.e., just before ATP removal). $V_m$ was −20 to −80 mV. (**B**) Relaxation time constants of the currents in (**A**), obtained by fits to single exponentials. Data are shown as mean ± standard error of the mean (SEM) from seven to nine experiments using a logarithmic ordinate. (**C**) Thermodynamic mutant cycle showing mutation-induced changes in the height of the free enthalpy barrier for the $B_1 \rightarrow IB_1$ transition ($\Delta\Delta G^0_{T^\ddagger - B1}$, *numbers on arrows*; $k$, Boltzmann's constant; $T$, absolute temperature). Each corner is represented by the mutations introduced into positions Q1371 and G576 of E1371Q-hCFTR. $\Delta\Delta G_{int}(B_1 \rightarrow T^\ddagger)$ (*purple number*) is obtained as the difference between $\Delta\Delta G^0_{T^\ddagger - B1}$ values along two parallel sides of the cycle. (**D**) Currents of last open channels after ATP removal for the indicated human CFTR constructs (*color coded*). Recordings were done as in panel (**A**), but on patches with smaller numbers of channels. Membrane potential was −80 mV. *Black lines* indicate the analyzed segments. (**E**) Intraburst equilibrium constants obtained by dwell-time analysis for the four human CFTR constructs, plotted on a logarithmic scale. Data are shown as mean ± standard error of the mean (SEM) from five to six experiments. (**F**) Thermodynamic mutant cycle showing mutation-induced changes in the stability of the O state relative to the $C_f$ state ($\Delta\Delta G^0_{O-CF}$, numbers on arrows; $k$, Boltzmann's constant; $T$, absolute temperature). Each corner of the cycle is represented by the mutations introduced into positions Q1371 and G576 of E1371Q-hCFTR. $\Delta\Delta G_{int}(C_f \rightarrow O)$ (*purple number*) is obtained as the difference between $\Delta\Delta G^0_{O-CF}$ values along two parallel sides of the cycle.

reduced in any of the single or double mutants, for all of which $K_{eq|B}$ values remained within ~twofold (*Figure 3E*). Correspondingly, a mutant cycle built on $K_{eq|B}$ (*Figure 3F*) revealed an interaction energy $\Delta\Delta G_{int}(C_f \rightarrow O)$ of 0.44 ±0 .36 kT (*Figure 3F*, *purple*), which is not significantly different from zero (p = 0.29) suggesting that the strength of the interaction does not change significantly between the flickery closed and the open state.

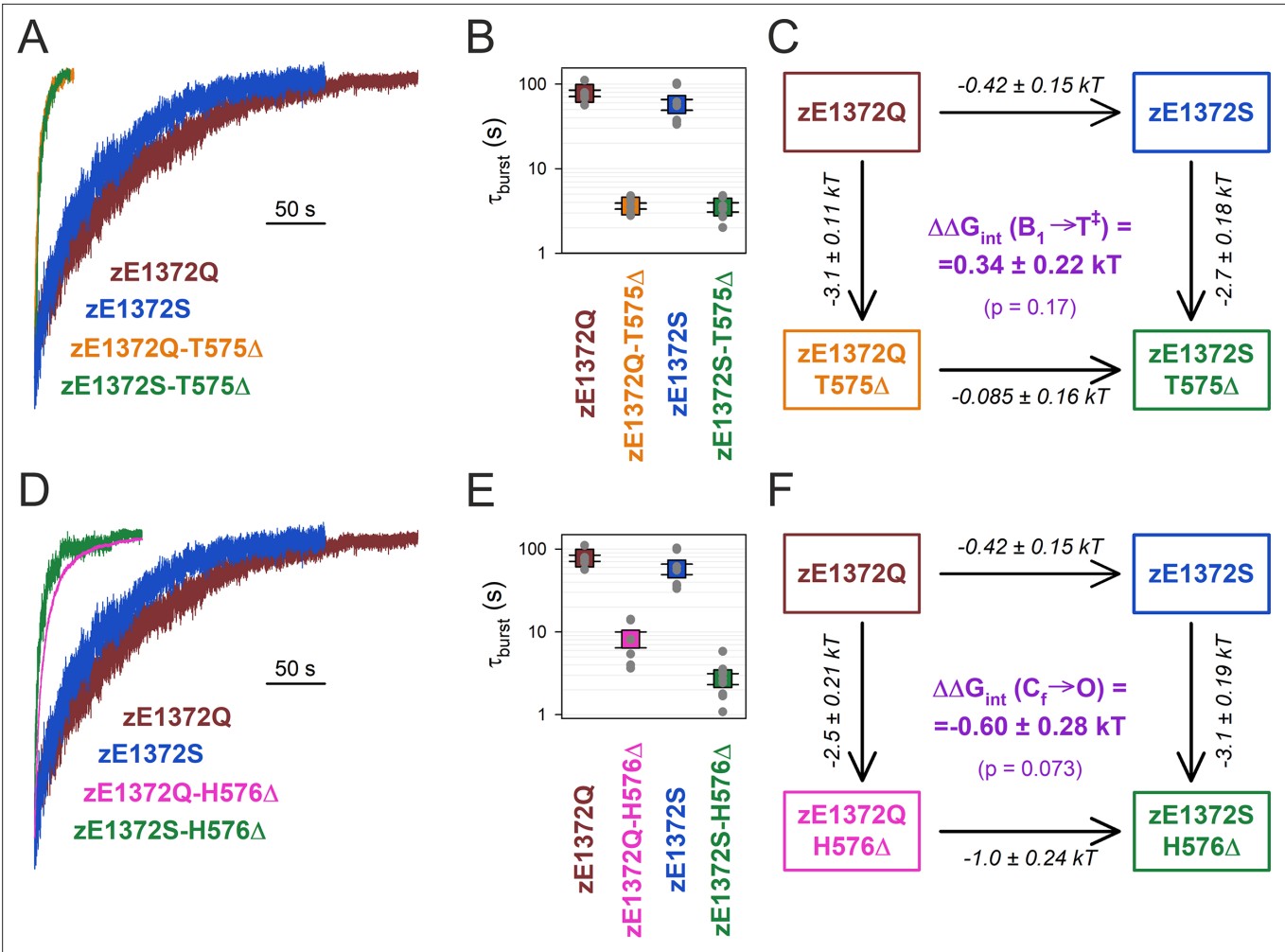

**Figure 4.** No coupling between positions 575/576 and 1372 of zCFTR E1372Q in the bursting state. (**A**, **D**) Macroscopic current relaxations following ATP removal for indicated zCFTR channel mutants (*color coded*). Experiments were performed as in *Figure 3A*, and current amplitudes are shown normalized by their steady-state values in ATP (i.e., just before ATP removal). (**B**, **E**) Relaxation time constants of the currents in (**A**, **D**), obtained by fits to single exponentials. Data are shown as mean ± standard error of the mean (SEM) from 6 to 10 experiments using a logarithmic ordinate. (**C**, **F**) Thermodynamic mutant cycles showing mutation-induced changes in the height of the free enthalpy barrier for the $B_1 \rightarrow IB_1$ transition ($\Delta\Delta G^0_{T^\ddagger-B1}$, *numbers on arrows; k*, Boltzmann's constant; *T*, absolute temperature). Each corner is represented by the mutations introduced into positions Q1372 and either T575 (**C**) or H576 (**F**) of E1372Q-zCFTR. $\Delta\Delta G_{int}(B_1 \rightarrow T^\ddagger)$ (*purple number*) is obtained as the difference between $\Delta\Delta G^0_{T^\ddagger-B1}$ values along two parallel sides of the cycle.

The online version of this article includes the following figure supplement(s) for figure 4:

**Figure supplement 1.** Close-up view of the region surrounding the mutated catalytic glutamate side chain (*brown sticks*) and D-loop residues zT575 (*orange sticks*) and zH576 (*magenta sticks*) in the cryo-EM structure of zCFTR-E1372Q, solved in the OF conformation (PDBID: 5w81).

## In E1372Q-zCFTR no gating state-dependent change in interaction strength is detectable between the zQ1372 side-chain amino group and the NBD1 D-loop backbone

To verify the lack of an analogous interaction in zCFTR, we tested whether the effects of NBD1 D-loop deletions on zCFTR non-hydrolytic closing rate depend on the presence of a glutamine side chain at position z1372 (*Figure 4*). Based on the uncertainty in the zQ1372 side-chain assignment (cf., *Figure 2F*) we constructed two mutant cycles, one built on deleting residue zT575 (*Figure 4A–C*), the other built on deleting residue zH576 (*Figure 4D–F*) from the NBD1 D-loop. Of note, by shortening the D-loop, both deletions are expected to increase the distance between the z1372 side chain and the D-loop backbone in general.

Interestingly, both D-loop deletions robustly destabilized the non-hydrolytic bursting state of zCFTR. However, that destabilization was largely independent of the nature of the side chain at position z1372, indicating that it is not caused by disruption of a zQ1372 side-chain interaction. Thus, deletion of zT575 shortened $\tau_b$ by ~20-fold, whether it was introduced into a zE1372Q (*Figure 4A*, *orange* vs. *brown trace*; *Figure 4B*, *orange* vs. *brown symbol*) or a zE1372S (*Figure 4A*, *green* vs. *blue trace*; *Figure 4B*, *green* vs. *blue symbol*) background. The coupling energy calculated from the resulting mutant cycle (*Figure 4C*), $\Delta\Delta G_{int}(B_1 \rightarrow T^\ddagger) = 0.34 \pm 0.22$ kT, is not significantly different from zero (p = 0.17). Similarly, deletion of zH576 shortened $\tau_b$ by ~10-fold in the zE1372Q background (*Figure 4D*, *magenta* vs. *brown trace*; *Figure 4E*, *magenta* vs. *brown symbol*), and by ~20-fold in the zE1372S background (*Figure 4D*, *green* vs. *blue trace*; *Figure 4E*, *green* vs. *blue symbol*), resulting in a calculated coupling energy (*Figure 4F*) of $-0.60 \pm 0.28$ kT, again not significantly different from zero (p = 0.073).

One possible explanation for the destabilizing effect of NBD1 D-loop deletions in zCFTR is the perturbation of a H-bond formed between the zT575 side chain and the zA1375 backbone, visible in the OF zCFTR structure (5w81; *Figure 4—figure supplement 1*). However, further investigation of that possibility is beyond the scope of the current study.

## Catalytic glutamate mutation E1371S and Walker A mutation K1250A both completely abolish ATP hydrolysis in hCFTR

In WT hCFTR channels the side chain of E1371 acts as the general base that deprotonates the attacking water molecule during the ATP hydrolysis reaction, whereas the side chain of the NBD2 Walker A lysine K1250 stabilizes the phosphate backbone of ATP bound in site 2, by forming strong ionic interactions with the β and γ phosphates (*Figure 5A*). The hK1250A and the hE1371S mutations both reduce channel closing rate by ~100-fold compared to WT hCFTR, indicating that they both reduce the rate of ATP hydrolysis ($k_1$) by at least 100-fold. Given the different roles the two side chains play in catalysis, if both single mutations failed to completely abolish $k_1$ then a further slowing in $k_1$, that is, a further prolongation of $\tau_b$, would be expected in the double mutant hK1250A/hE1371S. However, introducing the hE1371S mutation into the hK1250A background did not significantly alter $\tau_b$ (p = 0.13) (*Figure 5B*, *blue* vs. *gray trace*, *Figure 5C*, *blue* vs. *gray symbol*). When viewed from a different angle, mutation hK1250A also fails to prolong $\tau_b$ in the hE1371S background (*Figure 5B, C*, *blue* vs. *Figure 3A, B*, *blue*); instead it caused a slight (~28%) but insignificant (p = 0.052) shortening. These results together indicate that ATP hydrolysis is already completely abolished in both single mutants.

## Walker B mutant D1370N completely abolishes ATP hydrolysis in hCFTR

The conserved NBD2 Walker B aspartate hD1370 coordinates the $Mg^{2+}$ ion in site 2, required for ATP hydrolysis in all ATPases (*Figure 5A*). Although in other ABC proteins D-to-N mutations at the analogous position were shown to abrogate ATP hydrolysis (*Urbatsch et al., 1998*; *Hrycyna et al., 1999*; *Rai et al., 2006*), for the hCFTR NBD2 Walker B mutant D1370N $\tau_b$ is only ~2 s, that is, ~15 times shorter than that of most other non-hydrolytic mutants. If that relatively fast closing rate of D1370N hCFTR simply reflected residual ATP-ase activity, that is, a rate $k_1$ of ~0.5 $s^{-1}$, then introducing the hE1371S mutation into the hD1370N background would be expected to abolish that residual activity, and prolong $\tau_b$ to a value near that of single mutant hE1371S. However, introducing the hE1371S mutation into the hD1370N background did not significantly alter $\tau_b$ (p = 0.09) (*Figure 5D*, *blue* vs. *gray trace*, *Figure 5E*, *blue* vs. *gray symbol*), suggesting complete lack of ATP hydrolysis in the hD1370N single mutant.

## The E1371Q mutation stabilizes the bursting state of all non-hydrolytic hCFTR mutants

If the hE1371Q mutation indeed introduces an artificial H-bond which selectively forms in the bursting state, then introduction of that mutation into any other non-hydrolytic background construct is expected to prolong $\tau_b$. To verify that prediction, we introduced mutation hE1371Q into both Walker mutant backgrounds. Indeed, introducing mutation hE1371Q into an hK1250A background further increased $\tau_b$ by ~fivefold (p = $2.4 \times 10^{-5}$) indicating strong stabilization of state $B_1$ by the hE1371Q mutation (*Figure 5B*, *black* vs. *gray trace*, *Figure 5C*, *black* vs. *gray symbol*). Similarly, introducing the

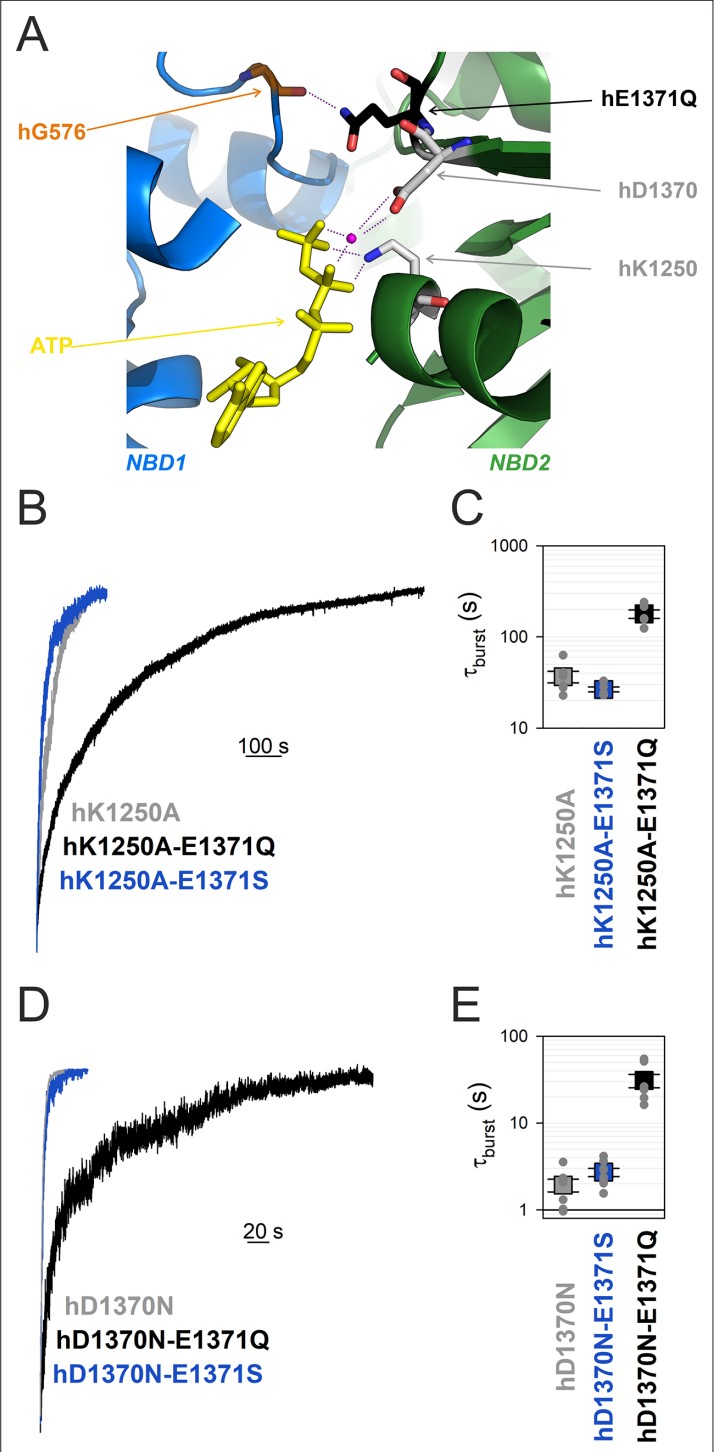

**Figure 5.** Mutation hE1371Q strongly stabilizes, whereas hD1370N destabilizes, bursts in other non-hydrolytic backgrounds. (**A**) Close-up view of the site 2 interface in the OF structure of hCFTR-E1371Q (PDBID: 6msm) highlighting (in sticks) NBD1 D-loop residue hG576 (*orange*), and NBD2 residues hQ1371 (*black*), hD1370 (*gray*), and hK1250 (*gray*). ATP, *yellow sticks*; Mg$^{2+}$, *purple sphere*. (**B**, **D**), Macroscopic current relaxations following ATP removal for indicated CFTR channel mutants (*color coded*). Experiments were performed as in *Figure 3A*, and current amplitudes are shown normalized by their steady-state values in ATP (i.e., just before ATP removal). (**C**, **E**), Relaxation time constants of the currents in (**B**, **D**), obtained by fits to single exponentials, displayed on a logarithmic ordinate. Data show mean ± standard error of the mean (SEM) from five to eight experiments.

The online version of this article includes the following figure supplement(s) for figure 5:

*Figure 5 continued on next page*

*Figure 5 continued*

**Figure supplement 1.** Intra-NBD2 movements associated with ATP binding and tight NBD dimerization.

---

hE1371Q mutation into an hD1370N background further increased $\tau_b$ by ~16-fold (p = $1.7 \times 10^{-4}$), again indicating strong stabilization of the $B_1$ state (*Figure 5D*, *black* vs. *gray trace*, *Figure 5E*, *black* vs. *gray symbol*).

## Discussion

In ABC proteins the E-to-Q mutation of the catalytic base, a conserved glutamate that follows the Walker B aspartate, abrogates ATP hydrolysis and traps isolated NBD proteins in stable dimer form (*Moody et al., 2002*; *Smith et al., 2002*; *Janas et al., 2003*). For that reason, E-to-Q mutants have been adopted as model systems in a multitude of structural studies, in an attempt to trap the protein in a prehydrolytic state. Indeed, under cryo-EM conditions in the presence of ATP full-length E-to-Q mutant ABC proteins yield OF structures in which ATP is seen intact, confirming lack of hydrolysis (*Liu et al., 2017*; *Johnson and Chen, 2018*; *Kim and Chen, 2018*; *Olsen et al., 2020*; *Song et al., 2021*; *Huang et al., 2023*). In hCFTR a first hint for a possible additional role, apart from simply disrupting ATP hydrolysis, of the engineered Q1371 side chain came from a recent study which compared non-hydrolytic closing rates of CFTR orthologs (*Simon and Csanády, 2023*). Intriguingly, whereas for hCFTR $\tau_b$ is >10-fold longer for the E-to-Q compared to the E-to-S mutant (*Figure 2A, B*, *black* vs. *dark blue*), for the zebrafish ortholog $\tau_b$ of the E-to-S and E-to-Q mutants is similar (*Figure 2A, B*, *brown* vs. *light blue*). Careful comparison of the dimer interface structures of OF hCFTR and zCFTR suggests the presence of a non-native H-bond in the human structure which forms between the side-chain amide nitrogen of hQ1371 and the backbone carbonyl group of hG576, located at a distance of ~3 Å in the opposing D-loop of NBD1 (*Figure 2E*). Such a bond cannot form in WT hCFTR, as the native glutamate side chain cannot act as a H-donor. Mutant cycle analysis indeed confirmed strong energetic coupling between positions 1371 and 576 in hCFTR/E1371Q throughout the bursting state (*Figure 3A–C*), maintained even during flickery closures (*Figure 3D–F*). Of note, based on our functional results *per se* it would remain possible that hQ1371 interacts with some other NBD1 D-loop residue. However, the structure (whose resolution in this region is particularly high) does not support any alternative interaction: the distance between all other D-loop functional groups and either polar group of the hQ1371 side chain is >4.2 Å. Interestingly, formation of that H-bond in the E-to-Q mutant seems a unique feature of hCFTR, since the analogous positions are further apart in OF structures of both zCFTR (4.3 Å) and of other ABCC proteins (e.g., 4.7 Å in bovine MRP1, PDBID: 6bhu).

In several ABC proteins Walker A K-to-A and Walker B D-to-N mutations were shown to decrease the rate of ATP hydrolysis below the limit of detection (*Ramjeesingh et al., 1999*; *Urbatsch et al., 1998*; *Hrycyna et al., 1999*). However, the sensitivities of such enzymatic assays are limited, and for hCFTR ATPase activity of the Walker B mutant D1370N has not been directly measured. Thus, for hCFTR the >10-fold faster closing rate of hK1250A and ~200-fold faster closing rate of hD1370N compared to hE1371Q raised the possibility of residual ATP hydrolysis in the Walker mutants. Here, we evaluated that possibility using combinations of catalytic site mutations. We found that mutation hE1371S does not significantly slow closing rate of either Walker mutant (*Figure 5B, C*, *Figure 5D, E*), confirming that all three mutations (hK1250A, hD1370N, and hE1371S) individually abolish ATP hydrolysis. Furthermore, the mutual lack of effect of mutation hE1371S on $\tau_b$ of hK1250A (*Figure 5B, C*, *blue* vs. *gray*) and of mutation hK1250A on $\tau_b$ of hE1371S CFTR (*Figure 5B, C*, *blue* vs. *Figure 3A, B*, *blue*) suggests that neither of those two mutations significantly alters the stability of the $B_1$ state. Thus, $\tau_b$ of hK1250A or hE1371S CFTR can be taken as relatively unbiased estimates of the life time of the $B_1$ state in WT hCFTR, suggesting a true $k_{-1}$ of ~0.03 s$^{-1}$ (*Figure 6*, *red*). Consistent with that notion, mutation hE1371S also failed to significantly alter $\tau_b$ of another non-hydrolytic background construct, hD1370N CFTR (*Figure 5D, E*, *blue* vs. *gray*).

In contrast, mutation hD1370N dramatically shortened $\tau_b$ of both hE1371S (by ~14-fold, *Figure 5D, E*, *blue* vs. *Figure 3A, B*, *blue*) and hE1371Q (by ~15-fold, *Figure 5D, E*, *black* vs. *Figure 3A, B*, *black*) CFTR, suggesting that the shorter $\tau_b$ of the hD1370N single mutant compared to other non-hydrolytic mutants is explained by a large destabilizing effect of the hD1370N mutation on $B_1$-state stability, as opposed to substantial residual ATP-ase activity. That conclusion is consistent with the monotonically

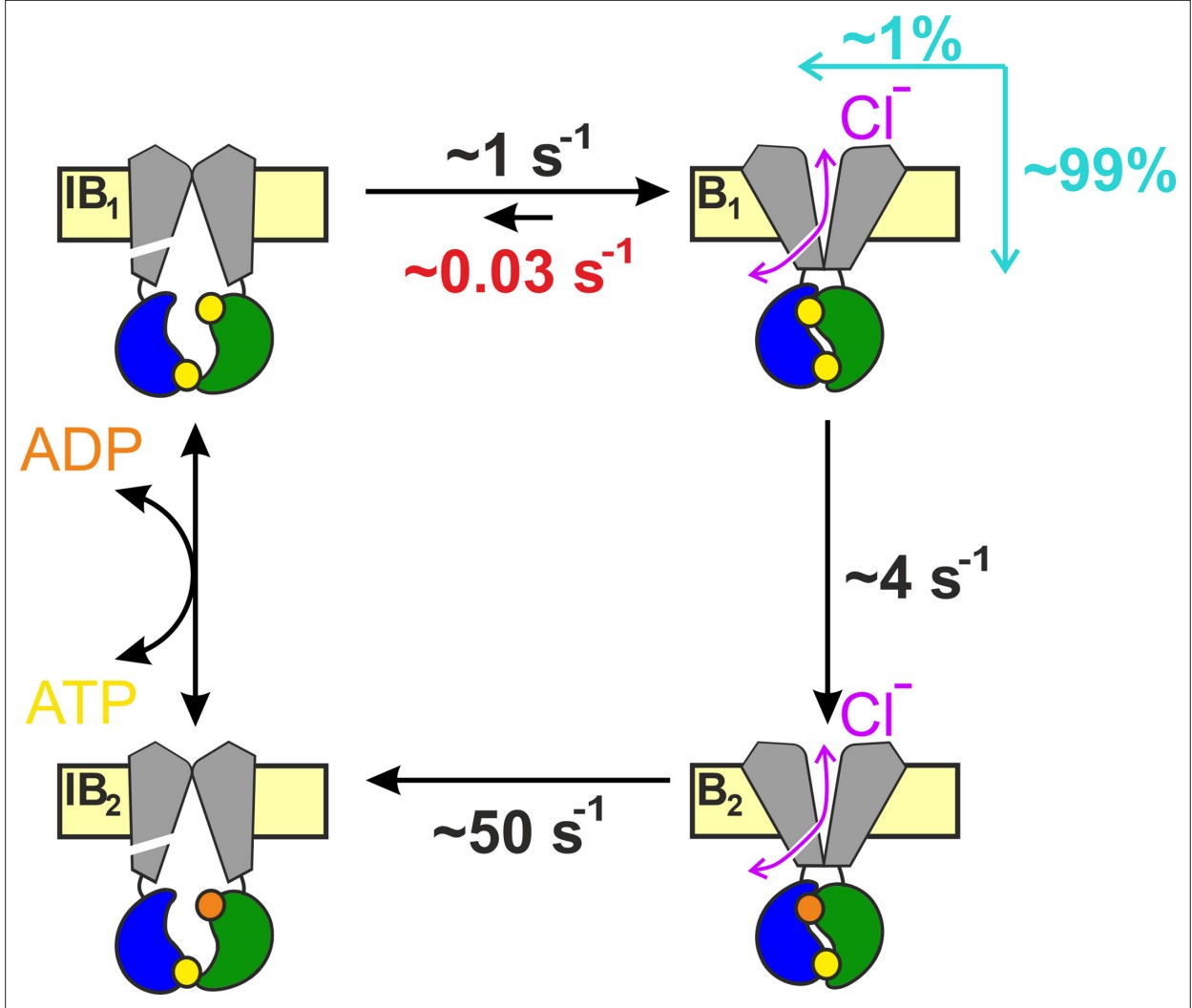

**Figure 6.** Coupling between bursts and ATP hydrolysis in wild-type (WT) hCFTR. Quantitative hCFTR gating cycle, color coding as in *Figure 1C*. Numbers represent estimated microscopic rate constants for prephosphorylated WT hCFTR channels gating in ATP at 25°C, with $k_{-1}$ (*red*) established in the present study. *Cyan arrows* and *numbers* illustrate the fractions of bursts that are terminated by ATP hydrolysis and non-hydrolytic dissociation of composite site 2, respectively.

decaying distribution of hD1370N burst durations, interpreted to reflect (near-) equilibrium gating of the mutant (*Csanády et al., 2010*).

Prompted by the discovery of the artificial H-bond in the hE1371Q mutant we sought to dissect its functional effects on $B_1$-state stability. Mutation hE1371Q indeed greatly increases $\tau_b$ when introduced into any other non-hydrolytic background: by ~13-fold in the hE1371S background (*Figure 3A, B*, *black* vs. *blue*), by ~16-fold in the hD1370N background (*Figure 5D, E*, *black* vs. *gray*), but only by ~5-fold in the hK1250A background (*Figure 5B, C*, *black* vs. *gray*). A possible explanation for the latter smaller effect could be that the K1250A mutation causes a slight increase in the hG576-hQ1371 distance, resulting in a small (~0.6 kT) but significant (p = 0.003) decrease in the strength of the non-native H-bond.

The observed shortening or lengthening of non-hydrolytic $\tau_b$ by different site 2 mutations is in line with similarly bidirectional effects of different mutations at the site 1 interface documented in earlier studies. Indeed, whereas non-hydrolytic $\tau_b$ is decreased ~5- to 10-fold by NBD1 Walker A mutation hK464A (*Powe et al., 2002*; *Vergani et al., 2003*; *Csanády et al., 2010*; *Csanády et al., 2013*), it is increased ~3-fold by NBD2 signature sequence mutation hH1348A (*Szollosi et al., 2011*; *Csanády et al., 2013*) or by $N^6$-(2-phenylethyl)-ATP (P-ATP) bound in site 1 (*Tsai et al., 2010*; *Csanády et al., 2013*), consistent with the notion that even site 1 undergoes substantial conformational changes

coincident with pore opening/closure (*Csanády et al., 2013*; *Mihályi et al., 2016*; *Sorum et al., 2017*; cf., cartoon in *Figure 1C*). Along those lines, it may seem possible that under conditions when ATP hydrolysis at site 2 is disrupted, pore closure might be initiated by separation of the NBD interface around site 1: if so, that would undermine the above interpretation of the lack of effect of the hK1250A mutation in an hE1371S background (and vice versa). However, our results directly demonstrate that separation around site 2 remains the prevalent pathway for pore closure even in the absence of ATP hydrolysis. If that were not the case then slowing of site 2 separation by the non-natural H bond would not be expected to prolong bursts. Specifically, if in hE1371S or hD1370N channels pore closure predominantly proceeded through dissociation of site 1 then further stabilization of site 2 by introduction of the E1371Q mutation would not be expected to prolong $\tau_b$ – in contrast to the ~13- to 16-fold prolongation observed in both backgrounds (*Figure 3A, B*, black vs. blue; *Figure 5D, E*, black vs. gray).

Earlier studies documented a clear pattern of larger effects on the rate of opening compared to non-hydrolytic closure ($k_{open}$ vs. $k_{-1}$, *Figure 1C*) for mutations introduced into opposing sides of site 2 (positions 555 [*Sorum et al., 2017*] and 1246 [*Sorum et al., 2015*]) in CFTR, suggesting that the inter-NBD contacts across the site 2 interface are already established in the transition state ($T^{\ddagger}$) that separates states $IB_1$ and $B_1$. The lack of effect on $k_{-1}$ of mutations hK1250A and hE1371S demonstrated here is in line with that suggestion. On the other hand, the ~15-fold increase and decrease in non-hydrolytic $\tau_b$ caused by mutations hE1371Q and hD1370N, respectively, clearly indicate that those two side chains (located near the center of the NBD dimer interface between sites 1 and 2; *Figure 5—figure supplement 1A*) undergo significant rearrangements even during the $T^{\ddagger} \leftrightarrow B_1$ gating step. Alignment of the cryo-EM structures of NBD2 in the absence of nucleotide (from unphosphorylated WT hCFTR apo structure 5uak), of NBD2-ATP without contact to NBD1 (from unphosphorylated WT ATP-bound hCFTR structure 8fzq), and of NBD2-ATP tightly dimerized with NBD1 (from phosphorylated ATP-bound hE1371Q structure 6msm) affords some speculation on the possible nature of those movements. For non-native hQ1371 that movement might involve flipping of the mutated side chain, prompted by the vicinity of the hG576 carbonyl oxygen established in state $T^{\ddagger}$. Indeed, whereas in the dimerized NBD2 structure the mutant hQ1371 side chain is clearly resolved and points toward the NBD1 D-loop (*Figure 2—figure supplement 1*), in the ATP bound but de-dimerized NBD2 structure no clear density is observable for the native E1371 side chain, suggesting that it is flexible or perhaps adopts a different rotameric conformation. For the hD1370 side chain the same two structures suggest a substantial rearrangement in the NBD-dimerized state which is not simply due to side-chain flipping: position 1370 is at the boundary between the 'head' and 'tail' subdomains of NBD2 which in all ABC proteins rotate toward each other upon ATP binding (*Karpowich et al., 2001*). Comparison of all three structures shows that in hCFTR's NBD2 that 'subdomain closure' is fully completed only in the NBD-dimerized state (*Figure 5—figure supplement 1B*), reminiscent to earlier findings on the maltose transporter (*Orelle et al., 2010*). That movement strengthens multiple interactions of the hD1370 side-chain carboxylate, by reducing its distance both to the $Mg^{2+}$ ion (from ~5.0 to 3.7 Å) and to the hS1251 side chain (from ~4.1 to 3.0 Å) (*Figure 5—figure supplement 1C*). If that movement happened during step $T^{\ddagger} \leftrightarrow B_1$ then lack of those tightening interactions in hD1370N could explain selective destabilization of the $B_1$ state relative to $T^{\ddagger}$ in the mutant.

In the past, assuming $k_{-1} < 0.2$ s$^{-1}$ and $k_1 \sim 4$ s$^{-1}$, a lower limit of ~0.95 was provided for the CR of prephosphorylated WT hCFTR gating at 25°C (*Csanády et al., 2010*). Our present estimate of $k_{-1} \sim 0.03$ s$^{-1}$ under the same conditions provides a more accurate value of CR ~0.99. Thus, only 1 out of a ~100 bursts is terminated by non-hydrolytic separation of site 2 (*Figure 6*). Furthermore, earlier estimates of temperature and PKA dependence of rates $k_1$ and $k_{-1}$ allow some extrapolations to different experimental conditions. Based on the measured activation enthalpies ($\Delta H^{\ddagger}_{B1 \rightarrow B2} \sim 70$ kJ/mol, $\Delta H^{\ddagger}_{B1 \rightarrow IB1} \sim 40$ kJ/mol; *Csanády et al., 2006*), $k_1$ is ~3-fold faster (~12 s$^{-1}$) and $k_{-1} \sim 1.9$-fold faster (~0.06 s$^{-1}$) at 37°C, predicting a CR of ~0.995. On the other hand, the presence of PKA slows both rates by ~twofold (*Csanády et al., 2010*; *Vergani et al., 2003*), predicting little effect on the CR.

A strong native H-bond between extracellular loops 1 and 6 of hCFTR (between residues R117 and E1124) was shown to be present only in the open state, but disrupted during both IB and flickery closures (*Simon and Csanády, 2021*). In contrast, we show here that the non-native hQ1371–hG576 H-bond is maintained during flickery closures, and disrupted only during IB closures (*Figure 3*). These findings are consistent with the notion that flickery closures of hE1371Q CFTR involve local movements

confined to the external ends of the TMD helices but no disruption of the tight NBD dimer, whereas IB closures represent a global rearrangement during which site 2 of the NBD dimer opens up to allow nucleotide exchange. The precise extent of that separation is presently unknown: it is large enough to disrupt the site 2 interfacial H-bonds (*Figure 5—figure supplement 1A*) formed between the side chains of hT1246 and hR555 in WT (or K1250R) hCFTR (*Vergani et al., 2005*) and between residues hQ1371 and hG576 in E1371Q hCFTR (*Figures 2E and 3A–C*), but too small to be detected by a pair of FRET sensors engineered into position h388 of NBD1 and position h1435 of NBD2 (*Levring et al., 2023*). Precisely gauging the extent of those motions will require a cryo-EM structure of the active IB state of phosphorylated hCFTR in the presence of ATP.

# Materials and methods

**Key resources table**

| Reagent type (species) or resource | Designation | Source or reference | Identifiers | Additional information |
|---|---|---|---|---|
| Biological sample (*Xenopus laevis*) | *Xenopus laevis* oocytes | European *Xenopus* Resource Centre | RRID: NXR_0.0080 | - |
| Commercial assay or kit | HiSpeed Plasmid Midi Kit | QIAGEN | 12643 | |
| Commercial assay or kit | QuickChange II Mutagenesis Kit | Agilent Technologies | 200524 | |
| Commercial assay or kit | mMESSAGE mMACHINE T7 Transcription Kit | Thermo Fisher Scientific | AM1344 | |
| Chemical compound, drug | Collagenase type II | Thermo Fisher Scientific | 17101-015 | |
| Chemical compound, drug | Adenosine 5'-triphosphoribose magnesium (ATP) | Sigma-Aldrich | A9187 | |
| Software, algorithm | Pclamp9 | Molecular Devices | RRID: SCR_011323 | |

## Molecular biology

Mutations were introduced into the human CFTR/pGEMHE and zebrafish CFTR/pGEMHE coding sequences using the Agilent QuickChange II mutagenesis Kit. The entire coding sequences of all constructs were confirmed by automated sequencing (LGC Genomics GmbH). Plasmids were linearized using Nhe I HF (New England Biolabs) and transcribed in vitro (mMessage-mMachine T7 Kit, Agilent Technologies), and purified cRNA was stored at −80°C.

## Functional expression of human CFTR constructs in *Xenopus laevis* oocytes

*X. laevis* oocytes were removed from anesthetized frogs following Institutional Animal Care and Use Committee guidelines, digested with collagenase (Gibco, Collagenase type II), and stored at 18°C in a solution containing (in mM) 82 NaCl, 2 KCl, 1 MgCl$_2$, 1.8 CaCl$_2$, and 5 4-(2-hydroxyethyl)-1-piperazineethanesulfonic acid (HEPES) (pH 7.5 with NaOH), supplemented with 50 µg/ml gentamycin. CFTR cRNA (1–20 ng) was injected in a fixed 50 nl volume (Drummond Nanoject II). Recordings were done 1–3 days following injection.

## Excised inside-out patch-clamp recording

Excised inside-out patch-clamp recordings were done as described (*Simon and Csanády, 2023*). The patch pipette solution contained (in mM): 138 N-Methyl-D-glucamine (NMDG), 2 MgCl$_2$, 5 HEPES, pH = 7.4 with HCl. The bath solution contained (in mM): 138 NMDG, 2 MgCl$_2$, 5 HEPES, 0.5 ethylene glycol-bis(β-aminoethyl ether)-N,N,N′,N′-tetraacetic acid (EGTA), pH = 7.1 with HCl. Following excision patches were placed into a flow chamber in which the continuously flowing bath solution could be exchanged with a time constant of <100 ms using electronic valves (ALA-VM8, Ala Scientific Instruments). Recordings were performed at 25°C, at membrane potentials of −20 to −80 mV. (-80 mV was used for patches with smaller numbers of channels, to facilitate intraburst kinetic analysis of last-channel segments; −20 to −40 mV was used when recording large currents, to increase seal stability; of note, CFTR channel gating is largely voltage independent (*Cai et al., 2003*; *Csanády and Töröcsik, 2014*).) MgATP (2–10 mM, Sigma-Aldrich) was diluted from a 400-mM aqueous stock

solution. Channels were activated by ~2-min exposure to 300 nM bovine PKA catalytic subunit, prepared from beaf heart following the protocol of *Kaczmarek et al., 1980*, in the presence of saturating ATP (2 mM for hE1371S/Q and zE1372S/Q backgrounds, 5 mM for hD1370N backgrounds, 10 mM for hK1250A backgrounds). Currents were amplified, low-pass filtered at 2 kHz (Axopatch 200B, Molecular Devices), digitized at 10 kHz (Digidata 1550B, Molecular Devices) and recorded to disk (Pclamp 11, Molecular Devices).

### Kinetic analysis of electrophysiological data

To obtain relaxation time constants ($\tau_{burst}$), macroscopic current relaxations were fitted to single exponentials (Clampfit 11). For intraburst kinetic analysis of the last open channel, recordings were Gaussian filtered at 100 Hz, idealized by half-amplitude threshold crossing, and mean open times ($\tau_{open}$) and mean flickery closed times ($\tau_{flicker}$) obtained as the simple arithmetic averages of the mean open and closed dwell-time durations, respectively. The intraburst equilibrium constant (*Figure 3E*) was calculated as $K_{eq|B} = \tau_{open}/\tau_{flicker}$.

### Mutant cycle analysis

Changes in the strength of specific residue–residue interactions between various gating states of hCFTR-E1371Q and of zCFTR-E1372Q were quantitated by mutant cycle analysis as described (*Mihályi et al., 2016*). In brief, mutation-induced changes in the height of the transition-state barrier for non-hydrolytic closure (step $B_1 \rightarrow T^{\ddagger}$) were calculated as $\Delta\Delta G^0_{T^{\ddagger}-B_1} = -kT \ln(r^{\lambda}/r)$, where $k$ is Boltzmann's constant, $T$ is absolute temperature, and $r$ and $r'$ are the rates for the $B_1 \rightarrow IB_1$ transition in the background construct and in the mutant, respectively, obtained as $1/\tau_{burst}$. The change in the stability of the O relative to the $C_f$ ground state was calculated as $\Delta\Delta G^0_{O-C_f} = -kT \ln(K^{\lambda}_{eq|B}/K_{eq|B})$, where $K_{eq|B}$ and $K^{\lambda}_{eq|B}$ are the equilibrium constants for the $C_f \leftrightarrow O$ transition in the background construct and in the mutant, respectively. Interaction free energy ($\Delta\Delta G_{int}$) was defined as the difference between $\Delta\Delta G^0$ values along two parallel sides of a mutant cycle. If the interaction between the target sites is completely abolished in each single mutant and in the double mutant, $\Delta\Delta G_{int}(X \rightarrow Y)$ quantifies the change in the strength of the target interaction in the background construct while the channel proceeds from state X to state Y (*Mihályi et al., 2016*). For the hQ1371–hG576 target interaction that assumption is met, as the only D-loop backbone carbonyl oxygen within 4.2 Å of the hQ1371 amino group is that of residue hG576, and both the hE1371S and hG576Δ single mutations are expected to further increase the distance between the D-loop backbone and the h1371 side chain, precluding formation of alternative interactions in the mutants. All $\Delta\Delta G$ values are given as mean ± standard error of the mean (SEM); SEM values were estimated assuming that $r$ and $K_{eq|B}$ are normally distributed random variables, using second-order approximations of the exact integrals (*Mihályi et al., 2016*).

### Statistics

All values are given as mean ± SEM, with the numbers of independent biological replicates provided in the figure legends. Significances were evaluated using Student's paired *t*-test.

### Acknowledgements

Supported by EU Horizon 2020 Research and Innovation Program grant 739593, Cystic Fibrosis Foundation Research Grant CSANAD21G0, and NKFIH KKP_22 grant 144199 to LC. MAS received support from the ÚNKP-22-3-II-SE-12 New National Excellence Program of the Ministry for Innovation and Technology from the source of the National Research, Development and Innovation Fund.

### Additional information

#### Competing interests

László Csanády: Reviewing editor, *eLife*. The other authors declare that no competing interests exist.

## Funding

| Funder | Grant reference number | Author |
|---|---|---|
| EU Horizon 2020 Research and Innovation Program | 739593 | László Csanády |
| Cystic Fibrosis Foundation | CSANAD21G0 | László Csanády |
| National Research, Development and Innovation Office | KKP 144199 | László Csanády |
| Ministry for Innovation and Technology | ÚNKP-22-3-II-SE-12 | Márton A Simon |

The funders had no role in study design, data collection, and interpretation, or the decision to submit the work for publication.

## Author contributions

Márton A Simon, Conceptualization, Data curation, Formal analysis, Funding acquisition, Investigation, Visualization, Writing – review and editing; Iordan Iordanov, Andras Szollosi, Resources; László Csanády, Conceptualization, Data curation, Supervision, Funding acquisition, Validation, Writing - original draft, Project administration, Writing – review and editing

## Author ORCIDs

Iordan Iordanov (b) http://orcid.org/0000-0001-8251-5857
Andras Szollosi (b) http://orcid.org/0000-0002-5570-4609
László Csanády (b) http://orcid.org/0000-0002-6547-5889

## Ethics

This study was performed in strict accordance with the recommendations in the Guide for the Care and Use of Laboratory Animals of the National Institutes of Health. The protocols were approved by the Institutional Animal Care and Use Committee of Semmelweis University (Assurance number: SEMMAWB/2023-001).

## Decision letter and Author response

Decision letter https://doi.org/10.7554/eLife.90736.sa1
Author response https://doi.org/10.7554/eLife.90736.sa2

---

# Additional files

## Supplementary files

• MDAR checklist

## Data availability

All data generated or analyzed during this study are included in the manuscript and the figures.

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
