## [Editor Report]

This important study provides a convincing mechanistic dissection of the contributions to gating and ATP hydrolysis from different CFTR channel mutations that have been exploited in structural studies to stabilize the open state of the channel. This is achieved by comparing structural information between channel orthologues from different species, by performing single-channel and macroscopic current recordings and cleverly combining different residue substitutions, and calculating mutant cycles. The precise quantitation of the effects of mutations in this study allows estimation of the energetics of a key transition in the channel gating cycle that had remained elusive. The findings are important for biophysicists and physiologists interested in CFTR channels and ABC transporters in general.

---

## [Decision Letter]

**Decision letter after peer review:**

Thank you for submitting your article "Estimating the true stability of the prehydrolytic outward-facing state in an ABC protein" for consideration by *eLife*. Your article has been reviewed by 3 peer reviewers, including Andres Jara-Oseguera as Reviewing Editor and Reviewer #1, and the evaluation has been overseen by Merritt Maduke as the Senior Editor.

The reviewers have discussed their reviews with one another, and the Reviewing Editor has drafted this to help you prepare a revised submission. All three reviewers agreed that the work presented is of high quality, and that the findings are relevant and interesting. However, reviewers also considered that some of the assumptions made throughout the manuscript and the limitations of the data should be more clearly acknowledged and discussed. Also, strong support for the stabilizing effect of the non-native hydrogen bond described requires some additional experimental work. These points that need to be addressed are described in the list of Essential Revisions below.

Essential revisions:

1) The authors hypothesized that in hCFTR an artificial H-bond between the side-chain of glutamine at position 1371 (i.e., in E1371Q mutant) and the backbone carbonyl at G576 of the D-loop stabilizes the NBD dimer. Such H-bond is absent in E1372Q zCFTR. The authors employed mutant cycle analysis on the G576Δ-E1371S mutation pair to demonstrate an energetic coupling between the hG576 and hE1371Q. However, how deletion of G576 might alter the local structure is unpredictable. The result does not directly address the discrepancy between zCFTR and hCFTR, either. The D-loop is highly conserved across species with a consensus sequence PFGYLD (residue 574-579 in hCFTR), but in zCFTR the analogous sequence is PFTHLD. The backbone carbonyl oxygen could therefore be harder to access in zCFTR.

Below we provide three alternative suggestions to provide additional support for the interpretation of the discrepancies between human and zebrafish (and other ABC transporters) regarding the presence of a stabilizing H bond involving a catalytic glutamine and the D-loop. The authors can also decide to take a different strategy to provide one more piece of evidence for this point.

– Introducing mutation in the D-loop of zCFTR to match the sequence of hCFTR (and vice versa). The authors' hypothesis would predict that zCFTR with hCFTR's D-loop sequence should recapitulate hCFTR's phenotype: the E-to-Q mutation on the catalytic glutamate would further lengthen the burst duration compared to the E-to-S mutation.

– The mutant cycle could be further applied to the zebrafish, by measuring the closing rate of a zebrafish construct containing the zE1372Q mutation together with a deletion on the D-loop. If D-loop deletion does not give a significant burst shortening in the zE1372Q background, it would support the idea that the H-bond in the hE1371Q-CFTR is unique.

– Obtain analysis of molecular dynamics simulations that shows that there is a hydrogen bond in the human simulations and not the zebrafish. Several papers have been published in which extensive simulations were carried out.

doi.org/10.1007/s00018-019-03211-4

doi.org/10.1016/j.bpj.2018.03.003

doi.org/10.1007/s00018-022-04621-7

Most studies focused their analysis on the transmembrane domains, but results will also include information on the NBDs. It might be possible for Prof Csanády to reach out to some of these computational groups and ask them to investigate the possible presence and persistence of a hydrogen bond between the glutamine and the NBD1 D-loop, in 6MSM or in 5W81.

2) Structurally, hG576, is better aligned to zT575. These two residues are almost completely overlapping in the backbone. There is a clear difference between the two pdb coordinates in the glutamine conformation, with the amide and carbonyl group positions swapped (Figure 2E-F): hQ1371 points the amide and zQ1372 points the oxygen towards the D loop, such that the zQ1372 amide group ends up closer to the backbone oxygen of zH576. The zQ1372 side chain is also shifted (~0.1 nm) closer to the γ phosphate, away from site 1, and further from the switch histidine (zH1403). Finally, site 1 is more open in zCFTR.

However, there are some uncertainties in modelling atomic coordinates from electron density maps. Negatively charged moieties often do not show up in these maps, for unknown reasons. In the electron density map for hCFTR (Figure Suppl. 1), part of the glutamine side chain (apparently the amide group) is missing. The electron density for both the amide and the carbonyl group is missing in the map for 5W81. How confident are we that the different pdb coordinates (hQ1371 vs. zQ1372) reflect a true experimental difference, with the glutamines in the bursting states taking on a different ensemble of conformations, such that only the human CFTR is stabilized by the hydrogen bond? Would it perhaps be possible that some other interaction involving the glutamine and the D-loop is also involved? This needs to be discussed.

3) The authors speculated that the reason for D1370N's relatively fast closing rate compared to other non-hydrolytic mutants is the loss of interaction between Mg^2+^ and the negatively charged aspartate. However, this reasoning fails to explain why non-hydrolytic closure of wildtype CFTR in the absence of Mg^2+^ (e.g., Levring et al. 2023 Extended Data Figure 7g) is even slower than the non-hydrolytic closure of D1370N CFTR opened by MgATP, where at least the Mg^2+^ is present. The authors should caution the readers that so far no definitive experimental evidence can explain the destabilizing effect of D1370N.

4) Based on the results that the double mutant E1371S/K1250A hCFTR has a similar burst duration as single mutant E1371S and K1250A, the authors made a strong claim that both mutations completely abolish ATP hydrolysis. Similar reasoning was applied to D1370N. The limitations in such interpretations should be discussed. The authors made the assumption that the termination of a burst is solely controlled by site 2 (Figure 1C). However, when hydrolysis is significantly diminished, binding of ATP in site 2 is very stable, and thus dissociation of ATP from site 2 versus site 1 becomes hard to distinguish. Whether all hydrolysis-deficient mutants share the same open-to-close transition by releasing ATP from site 2 but retaining ATP in site 1 is still a question. As the authors have elaborated in the text, it is known that mutations in the degenerate site 1 can affect non-hydrolytic closing. When mutations are introduced to site 2, they might as well result in allosteric effects on the stability of ATP binding in site 1, which could subsequently alter the channel's closing rate. The authors might want to make the readers aware of the complicated relationship between channel closure and CFTR's two ATP binding sites.

5) It is known that non-hydrolytic closing rate of CFTR is phosphorylation dependent, which the authors briefly mentioned in the Discussion. Vergani et al. (2003) documented that τburst of K1250A and D1370N in PKA is ~80 s and ~4 s respectively, but both are reduced by roughly twofold when PKA was removed. In this study the burst durations of K1250A (~30 s, Figure 4C) and D1370N (~2 s, Figure 4E) indicate that these channels are not strongly phosphorylated. Similarly, the τburst of E1371S in PKA is over 100 s (Bompadre et al. 2005), significantly longer than that in the current study. Although it is unclear how a different degree of R domain phosphorylation affects non-hydrolytic closing, the fact that it does again suggests that the simplified scheme used as the base for data interpretation may have its limitation. The Discussion would benefit from a more cautionary note on the oversimplification of the IB1↔B1 transition, and clarify that channels are not strongly phosphorylated in the current experimental condition.

6) The τburst of E1371Q CFTR is over 400 second while the τburst of K1250A-E1371Q double mutant is shortened to ~200 second (Figure 3B, black vs Figure 4C, black). The K1250A-E1371S CFTR also seems to have a shorter τburst than E1371S CFTR (Figure 4C, blue vs Figure 3B, blue). Although the effect of the K1250A mutation on shortening τburst of E1371Q and E1371S CFTR is not as dramatic as the D1370N mutation, the authors might want to clearly state if there is indeed a significant difference and address how K1250A mutation has such destabilizing effect, and provide additional statistical analysis.

7) page 10 line 214: "If both single mutations failed to completely abolish k1 then a further slowing in k1, i.e., a further prolongation of taub, would be expected in the double mutant K1250A/E1371S."

This prediction assumes that the effects of the mutations on hydrolysis are independent. One could envisage scenarios in which approximately the same residual hydrolytic activity would remain in K1250A, E1371S and K1250A/E1371S, if the two mutations worked in a similar way to slow hydrolysis down. Admittedly this is extremely unlikely, as we know the Walker A lysine and Walker B glutamate play different roles in catalysis, as the authors explain just above. But since this is a crucial point in reaching one of the main conclusions of the paper, it might be worth developing the argument of the redundancy of the interaction more extensively.

8) It is unclear if the rates for different mutations were measured at -20 or -80 mV. This should be described, and if different voltages were used, a discussion should be provided for how the different voltages are accounted for in the energetic calculations from the rates.

9) A better description should be provided for how the decay currents were normalized.

10) Page 9, line 174 and following are very hard to follow. Possible modification: "The coupling energy (DDGint(B1→T‡); Figure 3C, purple) is a measure of the discrepancy in the effects caused by perturbation of one target position depending on whether the other target position is mutated or intact (Figure 3C, parallel arrows). DDGint(B1→T‡) is thus given by the difference between DDG 0T++ -B1 values on parallel sides of the cycle (Figure 3C, numbers on parallel arrows). DDGint(B1→T‡) thus quantifies the change in the strength of the G576-Q1371 interaction in the E1371Q background construct while the channel proceeds from the B1 state to state T‡."

The latter statement is true only if simplifying assumptions are met. Among these, it is assumed that in each single mutant and in the double mutant the interaction between the target sites is completely abolished. This assumption is, strictly speaking, met in the deletion mutants. However, the serine could well form new contacts with carbonyl oxygens in the D-loop that depend on the glycine deletion. The statement should be qualified and/or the assumptions should be stated.

Non-essential revisions suggested by reviewers:

1) Including the traces for single mutants E1371Q and E1371S on Figure 4B-E would be helpful.

2) The different structures in Figure 4 Suppl. 1 are hard to distinguish, so it would be helpful to use more contrasting colors. It would also be valuable to include the loop for G576 for both structures on panel B.

3) Line 300-306: It would be helpful to provide a better clarification for the meaning of "bidirectional effects" – Does "bidirectional" here just mean both slowing down (as H1348A or P-ATP; E1371Q) or speeding up (as K464A; D1370N on k-1) of k-1, non-hydrolytic closure?

Also, the authors mentioned that non-hydrolytic closure is accelerated by K464A but slowed by H1348A and P-ATP, but it is unclear how perturbations in site 1 are related to the current scheme in this study, in which the IB1↔B1 transition considers only the separation of site 2. Some elaboration might be needed.

4) Figure 3D, E1371S: the scale for the traces appears different than from the other mutants. Is this due to increased overall noise in that recording?

5) The authors use IF, inward-facing, and OF, outward-facing, to refer to states in the kinetic scheme likely to adopt these conformations. Page 2, line 51 "The IF conformation, in which the pore is sealed near its extracellular end (Figure 1C, left), is a long lived state (~1 s) and corresponds to long "interburst" (IB) closed dwell times observable in single channel current recordings." I think it might be better to clearly separate structures and states at the start.

Possible alteration: "CFTR likely adopts an IF conformation, in which the pore is sealed near its extracellular end, during the long lived (~1 s) state (Figure 1C, left) which corresponds to long "interburst" (IB) closed dwell times observable in single channel current recordings. We refer to these closed states as "IF states"".

Then later: "When CFTR adopts the OF conformation the external gate is predominantly open, and a lateral portal which connects the channel pore with the cytosol generates a transmembrane aqueous pathway permeable to anions ((Zhang et al., 2018); Figure 1C, right, double arrows). However, functional studies show that the continuity of the transmembrane pore is occasionally disrupted for brief (~10 ms) intervals by a smaller conformational change (not depicted in Figure 1C) likely confined to the external ends of the TMD helices (Zhang et al., 2017; Zhang et al., 2018; Simon and Csanády, 2021).

Correspondingly, in single-channel recordings, this "OF" or "bursting" (B) state corresponds to clusters of channel openings separated by brief ("flickery") closures (Winter et al., 1994). The OF state is also relatively stable, with a dwell time of hundreds of milliseconds."

6) Page 4, line 80: the kinetic scheme used to describe CFTR (Figure 1) is used to illustrate ATP transporter ATPase cycles. Because the scheme has emerged from studies on CFTR gating, and not ABC exporter transport cycles (although the cycle might be similar), I would not refer to specific rates in the figure when introducing exporters. Possible alteration:

"The "coupling ratio" (CR), i.e., the fraction of initiated cycles that are completed through ATP hydrolysis, depends on the relative rates of the two possible exit pathways from the prehydrolytic state. In most ABC transporters the actual values of these rates are hard to directly estimate, but in CFTR direct measurements of conformational dwell times are made feasible by single-channel current recordings, providing estimates for microscopic transition rates. Thus, for a wild-type (WT) channel, we can estimate ATP hydrolysis rate (k1, Figure 1C) vs. non-hydrolytic NBD dimer dissociation rate (k-1, Figure 1C). Because typically k1>>k-1, CR=k1/(k1+k-1) is near unity."

7) Page 14, line 281 (and page 10 line 217): "the mutual lack of effect of mutation E1371S on taub of K1250A (Figure 4B-C, blue vs. gray) and of mutation K1250A on taub of E1371S CFTR" Why is introducing E1371S in K1250A described separately from introducing K1250A in E1371S? Since E1371S, K1250A, and K1250A/E1371S all have a similar taub it seems redundant to discuss the sequence in which the mutations are introduced. Why do the authors feel it is useful to present the results this way?

*Reviewer #1 (Recommendations for the authors):*

1. It is unclear if the rates for different mutations were measured at -20 or -80 mV. This should be described, and if different voltages were used, a discussion should be provided for how the different voltages are accounted for in the energetic calculations from the rates.

2. A better description should be provided for how the decay currents were normalized.

3. There is a confusing sentence on page 10, lines 217-218: "However, introducing the E1371S into the K1250A background did not significantly alter taub […]. Similarly, the introduction of mutation K1250A into the E1371S background failed to prolong taub." The two sentences say the same thing, because they refer to the same double mutant construct?

4. I suggest including the traces for single mutants E1371Q and E1371S in Figure 4B-E.

5. I suggest coloring the different structures in Figure 4 Suppl. 1 using more different colors – it is harder to see the differences with all shown in green. I would also suggest including the loop for G576 for both structures on panel B.

6. Although not necessary for publication, it would have been interesting and provided even stronger support for conclusions if the authors had performed the double mutant cycle analysis involving E1371Q and G576Δ in the background of one of the other non-hydrolytic mutations that have closing rates like WT, and potentially also the zebrafish channel, to show that the deletion of the residue at G576 has no effect on closure rate.

*Reviewer #2 (Recommendations for the authors):*

1. Line 300-306: What does "the bidirectional effects" mean for the studies on K464A, H1348A and P-ATP? The authors mentioned that non-hydrolytic closure is accelerated by K464A but slowed by H1348A and P-ATP, but it is unclear how perturbations in site 1 are related to the current scheme in this study, in which the IB1↔B1 transition considers only the separation of site 2. Some elaboration might be needed.

2. Figure 3D, E1371S: The scale is likely wrong. The single-channel amplitude, open-channel noise, and baseline noise are larger than the other three recordings. Please double check.

3. Figure 4-Figure Suppl. 1C: The color gradient of the three conformations could be increased to improve clarity. Currently the distinction between the three is somewhat hard to discern.

*Reviewer #3 (Recommendations for the authors):*

A. Suggestions for possible additional analysis.

While the electrophysiological study of the E to Q mutants is quite convincing (human and zebrafish E to Q mutants are clearly different, and there is some coupling with D-loop residues in hCFTR), I have some uncertainty regarding the interpretation offered.

Structurally, hG576, is better aligned to zT575. These two residues are almost completely overlapping in the backbone. There is a clear difference between the two pdb coordinates in the glutamine conformation, with the amide and carbonyl group positions swapped (Figure 2E-F): hQ1371 points the amide and zQ1372 points the oxygen towards the D loop, such that the zQ1372 amide group ends up closer to the backbone oxygen of zH576. The zQ1372 side chain is also shifted (~0.1 nm) closer to the γ phosphate, away from site 1, and further from the switch histidine (zH1403). Finally, site 1 is more open in zCFTR.

However, there are some uncertainties in modelling atomic coordinates from electron density maps. Negatively charged moieties often do not show up in these maps, for unknown reasons. In the electron density map for hCFTR (Figure Suppl. 1), part of the glutamine side chain (apparently the amide group) is missing. The electron density for both the amide and the carbonyl group are missing in the map for 5W81. How confident are we that the different pdb coordinates (hQ1371 vs. zQ1372) reflect a true experimental difference, with the glutamines in the bursting states taking on a different ensemble of conformations, such that only the human CFTR is stabilized by the hydrogen bond? Would it perhaps be possible that some other interaction involving the glutamine and the D-loop is also involved?

To strengthen the hypothesis of the different hydrogen bonding pattern the authors propose, one possible option is to look at molecular dynamics simulations. Several papers have been published in which extensive simulations were carried out.

doi.org/10.1007/s00018-019-03211-4

doi.org/10.1016/j.bpj.2018.03.003

doi.org/10.1007/s00018-022-04621-7

Most studies focused their analysis on the transmembrane domains, but results will also include information on the NBDs. It might be possible for Prof Csanády to reach out to some of these computational groups and ask them to investigate the possible presence and persistence of a hydrogen bond between the glutamine and the NBD1 D-loop, in 6MSM or in 5W81. I think some of these authors would be more than willing to collaborate.

B. General comment

This reader is fully convinced that the non-equilibrium gating model presented here realistically approximates CFTR gating. However, I think some readers might appreciate a more cautious presentation, avoiding all statements that might be read as giving for granted a given interpretation, and transparently and clearly describing the evidence underlying the most important assumptions. For instance, a bit more space in the introduction could be devoted to summarizing the evidence presented in reference Csanády et al., 2010, strongly supporting the kinetic scheme presented in Figure 1.

---

## [Author Response]

Essential revisions:1) The authors hypothesized that in hCFTR an artificial H-bond between the side-chain of glutamine at position 1371 (i.e., in E1371Q mutant) and the backbone carbonyl at G576 of the D-loop stabilizes the NBD dimer. Such H-bond is absent in E1372Q zCFTR. The authors employed mutant cycle analysis on the G576Δ-E1371S mutation pair to demonstrate an energetic coupling between the hG576 and hE1371Q. However, how deletion of G576 might alter the local structure is unpredictable.

We note that the modest effect of the G576 deletion observed in the E1371S background (Figure 3) makes it unlikely that the deletion should result in a major structural rearrangement.

The result does not directly address the discrepancy between zCFTR and hCFTR, either. The D-loop is highly conserved across species with a consensus sequence PFGYLD (residue 574-579 in hCFTR), but in zCFTR the analogous sequence is PFTHLD. The backbone carbonyl oxygen could therefore be harder to access in zCFTR.Below we provide three alternative suggestions to provide additional support for the interpretation of the discrepancies between human and zebrafish (and other ABC transporters) regarding the presence of a stabilizing H bond involving a catalytic glutamine and the D-loop. The authors can also decide to take a different strategy to provide one more piece of evidence for this point.– Introducing mutation in the D-loop of zCFTR to match the sequence of hCFTR (and vice versa). The authors' hypothesis would predict that zCFTR with hCFTR's D-loop sequence should recapitulate hCFTR's phenotype: the E-to-Q mutation on the catalytic glutamate would further lengthen the burst duration compared to the E-to-S mutation.

We do not think that such a prediction follows from our hypothesis. The suggested non-native H-bond in hCFTR involves a backbone carbonyl group in the NBD1 D-loop. The particular side chain patterns of the human and zebrafish D-loops have evolved in the context of the entire protein, and have been optimized to match the particular structures that surround them. It seems therefore highly unlikely that isolated replacement of the two divergent residues in zCFTR by those present in hCFTR should convert the conformation of the entire NBD interface region of zCFTR to precisely match that of hCFTR (or vice versa). E.g., our novel experiments on zCFTR NBD1 D-loop deletions (new Figure 4) indicate an important additional structural role of the zT575 side chain in zCFTR, which is clearly absent in hCFTR as the corresponding residue is a glycine (hG576).

– The mutant cycle could be further applied to the zebrafish, by measuring the closing rate of a zebrafish construct containing the zE1372Q mutation together with a deletion on the D-loop. If D-loop deletion does not give a significant burst shortening in the zE1372Q background, it would support the idea that the H-bond in the hE1371Q-CFTR is unique.

We have adopted this strategy and performed mutant cycle analyses in the zE1372Q background. We note upfront that the question relevant to our hypothesis is not whether an NBD1 D-loop deletion shortens non-hydrolytic bursts (D-loop residues might be involved also in other interactions that stabilize the burst!), but whether the effects of an NBD1 D-loop deletion depend on the presence of a glutamine at position zE1372.

In line with comment (2) below, we have constructed the mutant cycle in two different ways, based on deletion of either zT575 (new Figure 4A-C) or zH576 (new Figure 4D-F) in the zCFTR NBD1 D-loop. The results unequivocally demonstrate that the effects of either D-loop deletion are independent of whether a serine or a glutamine side chain is present at position z1372. Thus, ΔΔG_int_ for the zQ1372-D-loop interaction is not significantly different from zero (-0.60±0.28 kT (p=0.073) and 0.34±0.22 kT (p=0.17), respectively, based on the two models), in full support of our hypothesis.

Interestingly, in zCFTR both D-loop deletions strongly destabilize the burst by disrupting some other stabilizing interaction (possibly a H-bond between the side chain of zT575 and the backbone of zA1375; Figure 4 – Figure Suppl. 1). Further investigating this phenomenon is beyond the scope of the current study.

– Obtain analysis of molecular dynamics simulations that shows that there is a hydrogen bond in the human simulations and not the zebrafish.

In line with our expertise we have chosen the experimental approach discussed above.

2) Structurally, hG576, is better aligned to zT575. These two residues are almost completely overlapping in the backbone. There is a clear difference between the two pdb coordinates in the glutamine conformation, with the amide and carbonyl group positions swapped (Figure 2E-F): hQ1371 points the amide and zQ1372 points the oxygen towards the D loop, such that the zQ1372 amide group ends up closer to the backbone oxygen of zH576. The zQ1372 side chain is also shifted (~0.1 nm) closer to the γ phosphate, away from site 1, and further from the switch histidine (zH1403). Finally, site 1 is more open in zCFTR.However, there are some uncertainties in modelling atomic coordinates from electron density maps. Negatively charged moieties often do not show up in these maps, for unknown reasons. In the electron density map for hCFTR (Figure Suppl. 1), part of the glutamine side chain (apparently the amide group) is missing. The electron density for both the amide and the carbonyl group is missing in the map for 5W81. How confident are we that the different pdb coordinates (hQ1371 vs. zQ1372) reflect a true experimental difference,

This is always a valid concern, as the pdb coordinates are just those of a model. That is exactly why functional studies are needed to validate/disprove atomic details of model structures. Our functional results confirm that the positioning of the Q1371 side chain in the hCFTR model (6msm) is correct. We further note that in the zCFTR structure (5w81) the amide nitrogen of the zQ1372 side chain would be too far (3.72 Å) to interact with the zT575 backbone even if the positions of the zQ1372 amide and carbonyl groups were swapped (Figure 2F, *dark purple dotted line*). This is now discussed (lines 145-150), and our new experiments address both possible assignments (see response to comment 1), above.

Would it perhaps be possible that some other interaction involving the glutamine and the D-loop is also involved? This needs to be discussed.

Based on our functional results *per se* such a possibility can indeed not be excluded. However, the structure (whose resolution in this region is particularly high) does not support any alternative interaction: the distance between all other D-loop functional groups and either polar group of the hQ1371 side chain is >4.2 Å. This is now discussed in the text (lines 298-302).

3) The authors speculated that the reason for D1370N's relatively fast closing rate compared to other non-hydrolytic mutants is the loss of interaction between Mg^2+^ and the negatively charged aspartate. However, this reasoning fails to explain why non-hydrolytic closure of wildtype CFTR in the absence of Mg^2+^ (e.g., Levring et al. 2023 Extended Data Figure 7g) is even slower than the non-hydrolytic closure of D1370N CFTR opened by MgATP, where at least the Mg^2+^ is present. The authors should caution the readers that so far no definitive experimental evidence can explain the destabilizing effect of D1370N.

The cited study (Levring et al., 2023) does not present any data in conflict with our speculation, as neither the burst length of WT CFTR in the absence of Mg^2+^ nor that of D1370N were quantified. To our knowledge the only published study in which the effect of Mg^2+^ removal on burst length was quantified is that by Dousmanis and Gadsby (2002) on CFTR channels in guinea pig heart. Although absolute values of gating rates of those channels are not comparable to those of human epithelial CFTR, interestingly, guinea pig CFTR channels locked open by Mg^2+^ removal closed ~15-fold faster than those locked open by a non-hydrolyzable ATP analog.

Nevertheless, we do agree with the Reviewer that altered Mg^2+^ binding is not necessarily the sole reason for the shortened bursts of D1370N hCFTR. Indeed in the OF structure of WT hCFTR the D1370 side chain also strongly interacts with S1251, and that interaction is also incrementally strengthened as NBD2 proceeds through stepwise subdomain closure (5uak → 8fzq → 6msm). We have reworded this speculative proposal into a more general form (lines 378-382).

4) Based on the results that the double mutant E1371S/K1250A hCFTR has a similar burst duration as single mutant E1371S and K1250A, the authors made a strong claim that both mutations completely abolish ATP hydrolysis. Similar reasoning was applied to D1370N. The limitations in such interpretations should be discussed. The authors made the assumption that the termination of a burst is solely controlled by site 2 (Figure 1C). However, when hydrolysis is significantly diminished, binding of ATP in site 2 is very stable, and thus dissociation of ATP from site 2 versus site 1 becomes hard to distinguish. Whether all hydrolysis-deficient mutants share the same open-to-close transition by releasing ATP from site 2 but retaining ATP in site 1 is still a question.

We surmise that the Reviewer meant separation of the NBD interface around site 2 vs. site 1, not dissociation of ATP from site 2 versus site 1. (It is well established that pore gating is coupled to NBD interface movements rather than to nucleotide binding/unbinding events.)

In principle it might indeed happen that, under conditions when ATP hydrolysis at site 2 is disrupted, the NBD interface separates first around site 1. However, our results directly demonstrate that separation around site 2 remains the prevalent pathway for pore closure even in the absence of ATP hydrolysis. If that were not the case then further slowing of site 2 separation by the non-natural H bond would not be expected to prolong bursts. Specifically, if in hE1371S or hD1370N channels pore closure predominantly proceeded through dissociation of site 1 then further stabilization of site 2 by introduction of the E1371Q mutation would not be expected to prolong τ_b_ – in contrast to the ~13-16-fold prolongation observed in both backgrounds (Figure 3A-B, black vs. blue; Figure 5D-E, black vs. gray). This is now discussed in the text (lines 344-353).

As the authors have elaborated in the text, it is known that mutations in the degenerate site 1 can affect non-hydrolytic closing. When mutations are introduced to site 2, they might as well result in allosteric effects on the stability of ATP binding in site 1, which could subsequently alter the channel's closing rate. The authors might want to make the readers aware of the complicated relationship between channel closure and CFTR's two ATP binding sites.

Potential allosteric effects of site 2 mutations on site 1 stability cannot be ruled out in general. However, as explained above, our results strongly suggest that closing rate remains controlled by the separation of the site 2 interface, even in the mutants.

5) It is known that non-hydrolytic closing rate of CFTR is phosphorylation dependent, which the authors briefly mentioned in the Discussion.

We disagree. To put it more precisely, what is known is that non-hydrolytic closing rate depends on the presence or absence of PKA, i.e., that rate – just as the microscopic rates of all other gating steps – changes essentially instantaneously upon PKA removal in inside-out patches. On the other hand, there is strong evidence that the abrupt change in CFTR gating pattern upon PKA removal does not reflect dephosphorylation, but rather, loss of a direct effect of bound PKA on CFTR gating (Mihályi et al., 2020; PNAS 117: 21740–6).

Vergani et al. (2003) documented that τburst of K1250A and D1370N in PKA is ~80 s and ~4 s respectively, but both are reduced by roughly twofold when PKA was removed. In this study the burst durations of K1250A (~30 s, Figure 4C) and D1370N (~2 s, Figure 4E) indicate that these channels are not strongly phosphorylated.

We disagree. Just as in the study by Vergani et al. (2003), the patches in the current study were exposed to a high concentration (300 nM) of bovine PKA until the currents had reached steady state, and – as explained above – changes in gating following PKA removal are not caused by dephosphorylation (Mihályi et al., 2020). Thus, the channels in the current study are fully phosphorylated, and the differences in rates between those studies likely reflect modulation by PKA which remains bound until the channels close.

Similarly, the τburst of E1371S in PKA is over 100 s (Bompadre et al. 2005), significantly longer than that in the current study. Although it is unclear how a different degree of R domain phosphorylation affects non-hydrolytic closing, the fact that it does again suggests that the simplified scheme used as the base for data interpretation may have its limitation. The Discussion would benefit from a more cautionary note on the oversimplification of the IB1↔B1 transition, and clarify that channels are not strongly phosphorylated in the current experimental condition.

The fact that bound PKA modulates CFTR gating rates is briefly mentioned in the Discussion (lines 390-392), but the mechanism by which that happens is irrelevant to the scientific questions addressed here, and is beyond the scope of the present study. We agree that all kinetic schemes are "oversimplifications" of the systems they represent, as the microscopic rate constants typically depend on a multitude of factors (including temperature, pH, etc). Correspondingly, the legend to Figure 1C states that the gating scheme is "schematic", and that not all gating states are depicted. Nevertheless, we believe that it is sufficiently detailed to facilitate mechanistic interpretation of the observed phenomena.

6) The τburst of E1371Q CFTR is over 400 second while the τburst of K1250A-E1371Q double mutant is shortened to ~200 second (Figure 3B, black vs Figure 4C, black). The K1250A-E1371S CFTR also seems to have a shorter τburst than E1371S CFTR (Figure 4C, blue vs Figure 3B, blue). Although the effect of the K1250A mutation on shortening τburst of E1371Q and E1371S CFTR is not as dramatic as the D1370N mutation, the authors might want to clearly state if there is indeed a significant difference and address how K1250A mutation has such destabilizing effect, and provide additional statistical analysis.

In the E1371S background introduction of the K1250A mutation decreased τ_burst_ only by ~28%, which failed to reach significance (p=0.052). The exact number and the significance value are now provided in the text (line 251).

On the other hand, the ~2-fold shortening of τ_burst_ in the E1371Q background is indeed a significant effect (p=0.003). Another way to express the same fact is to state (see lines 329-332) that the increase in τ_burst_ caused by the E1371Q mutation is only ~5-fold in the K1250A background, as opposed to 13-16-fold in the hE1371S or hD1370N backgrounds. A possible explanation could be that the K1250A mutation causes a slight increase in the hG576-hQ1371 distance, resulting in a small (~0.6 kT) but significant (p=0.003) decrease in the strength of the non-native H-bond. This is now discussed in the text, and the significance value is provided (lines 332-335).

7) page 10 line 214: "If both single mutations failed to completely abolish k1 then a further slowing in k1, i.e., a further prolongation of taub, would be expected in the double mutant K1250A/E1371S."This prediction assumes that the effects of the mutations on hydrolysis are independent. One could envisage scenarios in which approximately the same residual hydrolytic activity would remain in K1250A, E1371S and K1250A/E1371S, if the two mutations worked in a similar way to slow hydrolysis down. Admittedly this is extremely unlikely, as we know the Walker A lysine and Walker B glutamate play different roles in catalysis, as the authors explain just above. But since this is a crucial point in reaching one of the main conclusions of the paper, it might be worth developing the argument of the redundancy of the interaction more extensively.

Thank you, argument added (lines 245-246).

8) It is unclear if the rates for different mutations were measured at -20 or -80 mV. This should be described, and if different voltages were used, a discussion should be provided for how the different voltages are accounted for in the energetic calculations from the rates.

CFTR channel gating is largely voltage-independent: P_o_ differs by only ~15% between -100 and +100 mV (Cai et al., 2003), and non-hydrolytic closing rates at -120 and +60 mV are similar (Csanády et al., 2014). Thus, the voltage is irrelevant for the calculation of closing rates which simply reflect the inverses of the fitted time constants (see line 461). In the present study non-hydrolytic closing rates were obtained between -20 and -80 mV; -80 mV was used for patches with smaller numbers of channels, to facilitate intraburst kinetic analysis of last-channel segments; -20 to -40 mV was used when recording large currents, to increase seal stability. This is now explained more clearly in Materials and methods (lines 435-438).

9) A better description should be provided for how the decay currents were normalized.

In each figure legend we have replaced the sentence "current amplitudes are shown rescaled by their steady-state values" with the more detailed statement "current amplitudes are shown normalized by their steady-state values in ATP (i.e., just before ATP removal)".

10) Page 9, line 174 and following are very hard to follow. Possible modification: "The coupling energy (DDGint(B1→T‡); Figure 3C, purple) is a measure of the discrepancy in the effects caused by perturbation of one target position depending on whether the other target position is mutated or intact (Figure 3C, parallel arrows). DDGint(B1→T‡) is thus given by the difference between DDG 0T++ -B1 values on parallel sides of the cycle (Figure 3C, numbers on parallel arrows). DDGint(B1→T‡) thus quantifies the change in the strength of the G576-Q1371 interaction in the E1371Q background construct while the channel proceeds from the B1 state to state T‡."

Corrected as suggested, thank you (lines 178-184).

The latter statement is true only if simplifying assumptions are met. Among these, it is assumed that in each single mutant and in the double mutant the interaction between the target sites is completely abolished. This assumption is, strictly speaking, met in the deletion mutants. However, the serine could well form new contacts with carbonyl oxygens in the D-loop that depend on the glycine deletion. The statement should be qualified and/or the assumptions should be stated.

This is highly unlikely. The only D-loop backbone carbonyl oxygen within 4.2 Å of the hQ1371 amino group is that of residue hG576, and the hE1371S mutation is expected to further increase the distance between the D-loop backbone and the h1371 side chain by ~2 Å. Thus, formation of alternative interactions between the D-loop and the hS1371 side chain is precluded. We now discuss the validity of these assumptions in Materials and methods (lines 465-472).

Non-essential revisions suggested by reviewers:1) Including the traces for single mutants E1371Q and E1371S on Figure 4B-E would be helpful.

We have decided not to add these: at the time scale required to plot the E1371Q trace the other traces would all become squashed on top of each other, especially the traces in panel D.

2) The different structures in Figure 4 Suppl. 1 are hard to distinguish, so it would be helpful to use more contrasting colors. It would also be valuable to include the loop for G576 for both structures on panel B.

This panel focuses on the environment of the D1370 side chain. Adding the D-loop would make the panel overcrowded and distract the attention from its message.

3) Line 300-306: It would be helpful to provide a better clarification for the meaning of "bidirectional effects" – Does "bidirectional" here just mean both slowing down (as H1348A or P-ATP; E1371Q) or speeding up (as K464A; D1370N on k-1) of k-1, non-hydrolytic closure?

Yes, that is correct. This sentence now reads:

"The observed shortening or lengthening of non-hydrolytic τ_b_ by different site-2 mutations" (line 336).

Also, the authors mentioned that non-hydrolytic closure is accelerated by K464A but slowed by H1348A and P-ATP, but it is unclear how perturbations in site 1 are related to the current scheme in this study, in which the IB1↔B1 transition considers only the separation of site 2. Some elaboration might be needed.

This is a misunderstanding. The fact that site 1 does not completely separate during pore opening/closure does not mean that it remains entirely static. Indeed, the observed effects on non-hydrolytic τ_b_ of various perturbations at the site-1 interface clearly demonstrate that even site 1 undergoes substantial conformational changes coincident with pore opening/closure, as has been elaborated in multiple earlier studies (Csanády et al., 2013; Mihályi et al., 2016; Sorum et al., 2017). Correspondingly, the cartoon in Figure 1C was drawn to illustrate some movement even in site 1. We have added a clarifying sentence (lines 342-344).

4) Figure 3D, E1371S: the scale for the traces appears different than from the other mutants. Is this due to increased overall noise in that recording?

Yes, the apparent difference is due to larger noise, the scaling is identical. No changes in conductance were observed for the E1371S construct.

5) The authors use IF, inward-facing, and OF, outward-facing, to refer to states in the kinetic scheme likely to adopt these conformations. Page 2, line 51 "The IF conformation, in which the pore is sealed near its extracellular end (Figure 1C, left), is a long lived state (~1 s) and corresponds to long "interburst" (IB) closed dwell times observable in single channel current recordings." I think it might be better to clearly separate structures and states at the start.Possible alteration: "CFTR likely adopts an IF conformation, in which the pore is sealed near its extracellular end, during the long lived (~1 s) state (Figure 1C, left) which corresponds to long "interburst" (IB) closed dwell times observable in single channel current recordings. We refer to these closed states as "IF states"".

Thank you, corrected as suggested (lines 51-54).

Then later: "When CFTR adopts the OF conformation the external gate is predominantly open, and a lateral portal which connects the channel pore with the cytosol generates a transmembrane aqueous pathway permeable to anions ((Zhang et al., 2018); Figure 1C, right, double arrows). However, functional studies show that the continuity of the transmembrane pore is occasionally disrupted for brief (~10 ms) intervals by a smaller conformational change (not depicted in Figure 1C) likely confined to the external ends of the TMD helices (Zhang et al., 2017; Zhang et al., 2018; Simon and Csanády, 2021).Correspondingly, in single-channel recordings, this "OF" or "bursting" (B) state corresponds to clusters of channel openings separated by brief ("flickery") closures (Winter et al., 1994). The OF state is also relatively stable, with a dwell time of hundreds of milliseconds."

Thank you, corrected as suggested (lines 54-62).

6) Page 4, line 80: the kinetic scheme used to describe CFTR (Figure 1) is used to illustrate ATP transporter ATPase cycles. Because the scheme has emerged from studies on CFTR gating, and not ABC exporter transport cycles (although the cycle might be similar), I would not refer to specific rates in the figure when introducing exporters. Possible alteration:"The "coupling ratio" (CR), i.e., the fraction of initiated cycles that are completed through ATP hydrolysis, depends on the relative rates of the two possible exit pathways from the prehydrolytic state. In most ABC transporters the actual values of these rates are hard to directly estimate, but in CFTR direct measurements of conformational dwell times are made feasible by single-channel current recordings, providing estimates for microscopic transition rates. Thus, for a wild-type (WT) channel, we can estimate ATP hydrolysis rate (k1, Figure 1C) vs. non-hydrolytic NBD dimer dissociation rate (k-1, Figure 1C). Because typically k1>>k-1, CR=k1/(k1+k-1) is near unity."

Thank you, corrected as suggested (lines 85-87).

7) Page 14, line 281 (and page 10 line 217): "the mutual lack of effect of mutation E1371S on taub of K1250A (Figure 4B-C, blue vs. gray) and of mutation K1250A on taub of E1371S CFTR" Why is introducing E1371S in K1250A described separately from introducing K1250A in E1371S? Since E1371S, K1250A, and K1250A/E1371S all have a similar taub it seems redundant to discuss the sequence in which the mutations are introduced. Why do the authors feel it is useful to present the results this way?

We discuss the three mutants in this sequence for didactic reasons. In principle, the effects of the two mutations might be asymmetric. E.g., if both mutations disrupted ATP hydrolysis, but only mutation K1250A affected NBD dimer stability, then introducing mutation K1250A into an E1371S background would affect τ_b_, whereas introducing mutation E1371S into a K1250A background would not. The mutual lack of (significant) effect thus shows that neither mutation (significantly) alters NBD dimer stability. We have reworded this sentence to read:

"When viewed from a different angle, mutation hK1250A also fails to prolong τ_b_ in the hE1371S background".